# Harmonising the land-use flux estimates of global models and national inventories for 2000-2020

Giacomo Grassi[1], Clemens Schwingshackl[2], Thomas Gasser[3], Richard A. Houghton[4], Stephen Sitch[5], Josep G. Canadell[6], Alessandro Cescatti[1], Philippe Ciais[7], Sandro Federici[8], Pierre Friedlingstein[9,10], Werner A. Kurz[11], Maria J. Sanz Sanchez[12,13], Raúl Abad Viñas[1], Ramdane Alkama[1], Selma Bultan[2], Guido Ceccherini[1], Stefanie Falk[2], Etsushi Kato[14], Daniel Kennedy[15], Jürgen Knauer[16], Anu Korosuo[1], Joana Melo[1], Matthew J. McGrath[7], Julia E.M.S. Nabel[17,18], Benjamin Poulter[19], Anna Romanovskaya[20], Simone Rossi[21], Hanqin Tian[22], Anthony P. Walker[23], Wenping Yuan[24], Xu Yue[25], Julia Pongratz[2,17]

[1] Joint Research Centre, European Commission, Ispra, Italy.

[2] Ludwig-Maximilians-Universität München, Munich, Germany.

[3] International Institute for Applied Systems Analysis (IIASA), Laxenburg, Austria.

[4] Woodwell Climate Research Center, Falmouth, MA, USA.

[5] Department of Geography, College of Life and Environmental Sciences, University of Exeter, Exeter, UK.

[6] Global Carbon Project, CSIRO Environment, Canberra, ACT, Australia.

[7] Laboratoire des Sciences du Climat et de l'Environnement CEA, CNRS, UVSQ, 91191 Gif sur Yvette, France

[8] Institute for Global Environmental Strategies, Hayama, Japan.

[9] College of Engineering, Mathematics and Physical Sciences, University of Exeter, Exeter, UK.

[10] Laboratoire de Météorologie Dynamique/Institut Pierre-Simon Laplace, CNRS, Ecole Normale.

[11] Canadian Forest Service, Natural Resources Canada, Victoria, British Columbia, Canada.

[12] Basque Centre for Climate Change (BC3), Sede Building, 1, 1st floor, Scientific Campus of the University of the Basque Country, 48940, Leioa, Spain.

[13] Ikerbasque, Basque Science Foundation, María Díaz Haroko Kalea, 3, 48013, Bilbo, Spain

[14] Institute of Applied Energy, Tokyo 105-0003, Japan.

[15] National Center for Atmospheric Research, Boulder, CO, USA.

[16] Hawkesbury Institute for the Environment, Western Sydney University, Penrith, NSW, Australia.

[17] Max Planck Institute for Meteorology, 20146 Hamburg, Germany.

[18] Max Planck Institute for Biogeochemistry, Jena, Germany.

[19] NASA Goddard Space Flight Center, Biospheric Sciences Laboratory, Greenbelt, MD 20771, USA.

[20] Yu. A. Izrael Institute of Global Climate and Ecology, Glebovskaya 20B, Moscow, Russia 107258

[21] Independent researcher: Celle Ligure, Italy

[22] Hanqin Tian, Schiller Institute for Integrated Science and Society, Department of Earth and Environmental Sciences, Boston College, Chestnut Hill, MA 02467, USA

[23] Environmental Sciences Division and Climate Change Science Institute, Oak Ridge National Laboratory, Oak Ridge, TN, 37831, USA.

[24] School of Atmospheric Sciences, Sun Yat-sen University, Zhuhai, China.

[25] School of Environmental Science and Engineering, Nanjing University of Information Science & Technology (NUIST), Nanjing, 210044, China.

*Correspondence to*: Giacomo Grassi (giacomo.grassi@ec.europa.eu)

**Abstract**

As the focus of climate policy shifts from pledges to implementation, there is a growing need to track progress on climate change mitigation at the country level, particularly for the land-use sector. Despite new tools and models providing unprecedented monitoring opportunities, striking differences remain in estimations of anthropogenic land-use $CO_2$ fluxes between the national greenhouse gas inventories (NGHGIs) used to assess compliance with national climate targets under the Paris Agreement, and the Global Carbon Budget and IPCC assessment reports, both based on global bookkeeping models (BMs).

Recent studies have shown that these differences are mainly due to inconsistent definitions of anthropogenic $CO_2$ fluxes in managed forests. Countries assume larger areas of forest to be managed than BMs due to a broader definition of managed land in NGHGIs. Additionally, the fraction of the land sink caused by indirect effects of human-induced environmental change (e.g., fertilization effect on vegetation growth due to increased atmospheric $CO_2$ concentration) on managed lands is treated as non-anthropogenic by BMs, but as anthropogenic in most NGHGIs.

Building on previous studies, we implement an approach that adds the $CO_2$ sink caused by environmental change in countries' managed forests (estimated by sixteen Dynamic Global Vegetation Models, DGVMs) to the land-use fluxes from three BMs. This sum is conceptually more comparable to NGHGIs and is thus expected to be quantitatively more similar. Our analysis uses updated and more comprehensive data from NGHGIs than previous studies and provides model results at a greater level of disaggregation in terms of regions, countries and land categories (i.e., forest land, deforestation, organic soils, other land uses).

Our results confirm a large difference (6.7 GtCO$_2$ yr$^{-1}$) in global land-use $CO_2$ fluxes between the ensemble mean of the BMs, which estimate a source of 4.8 GtCO$_2$ yr$^{-1}$ for the period 2000-2020, and NGHGIs, which estimate a sink of -1.9 GtCO$_2$ yr$^{-1}$ in the same period. Most of the gap is found on forest land (3.5 GtCO$_2$ yr$^{-1}$), with differences also for deforestation (2.4 GtCO$_2$ yr$^{-1}$), fluxes from other land uses (1.0 GtCO$_2$ yr$^{-1}$), and to a lesser extent for fluxes from organic soils (0.2 GtCO$_2$ yr$^{-1}$). By adding the DGVM ensemble mean sink arising from environmental change in managed forests (-6.4 GtCO$_2$ yr$^{-1}$) to BM estimates, the gap between BMs and NGHGIs becomes substantially smaller both globally (residual gap: 0.3 GtCO$_2$ yr$^{-1}$) and in most regions and countries. However, some discrepancies remain and deserve further investigation. For example, the BMs generally provide higher emissions from deforestation than NGHGIs and, when adjusted with the sink in managed forests estimated by DGVMs, yield a sink that is often greater than NGHGIs.

In summary, this study provides a blueprint for harmonising the estimations of anthropogenic land-use fluxes, allowing for detailed comparisons between global models and national inventories at global, regional, and country levels. This is crucial to increase confidence in land-use emissions estimates, support investments in land-based mitigation strategies and assess the countries' collective progress under the Global Stocktake of the Paris Agreement.

Data from this study are openly available via the Zenodo portal (Grassi et al., 2023), at https://zenodo.org/record/7541525#.Y8WF8ezMJEI

**Introduction**

In recent years, most countries have set new or revised goals for reducing emissions by 2030 (UNFCCC 2022a, Meinshausen et al. 2022). The focus of climate policy is now shifting towards the implementation of these goals, leading to greater interest in tracking progress at the country level. One sector of particular concern is land use, land-use change, and forestry (LULUCF), which accounts for 25% of emission reductions pledged by countries in their National Determined Contributions (NDCs) (Grassi et al., 2017). Reducing emissions from LULUCF is crucial for mitigating climate change (Roe et al., 2021; IPCC, 2022) and is receiving increasing attention at policy level. For example, at the United Nations Framework Convention on Climate Change (UNFCCC) conference of the Parties in Glasgow in 2021, 141 countries committed to ending forest loss and land degradation by 2030 (Taylor et al., 2021; Nabuurs et al., 2022; Gasser et al. 2022). However, monitoring and assessing progress in the LULUCF sector is difficult due to its complexity and the challenges of measuring greenhouse gas (GHG) emissions. Despite unprecedented monitoring opportunities offered by new observation platforms, tools, and models, striking differences remain between land-use $CO_2$ fluxes estimated by different approaches.

The Global Carbon Budget (Friedlingstein et al., 2022) employs Dynamic Global Vegetation Models (DGVMs) and global bookkeeping models (BMs) to estimate natural and anthropogenic historical $CO_2$ fluxes from land, respectively. These estimates are also used in the assessment reports of the Intergovernmental Panel on Climate Change (IPCC) (Canadell et al., 2021; IPCC, 2022). DGVMs estimate that terrestrial ecosystems absorb nearly 30% of the total anthropogenic $CO_2$ emissions (Friedlingstein et al. 2022), mainly in forests. BMs estimate that land use is a net source of $CO_2$ globally, mainly due to deforestation, equal to around 11% of total global anthropogenic $CO_2$ emissions in the last decade (Friedlingstein et al., 2022). However, the national greenhouse gas inventories (NGHGIs) used to assess compliance with the NDCs under the Paris Agreement report a net anthropogenic sink of $CO_2$ for the LULUCF sector globally (Grassi et al., 2022). This discrepancy between estimates from global models (BMs and DGVMs) and NGHGIs is confusing for policymakers, and makes the estimates appear contradictory.

Due to differences in purpose and scope, the largely independent scientific communities that support the IPCC Guidelines (reflected in NGHGIs) and those that support the IPCC assessment reports (based on global models) have developed different approaches to identify anthropogenic GHG fluxes, as illustrated in Figure 1. Previous studies (Grassi et al., 2018; 2021) suggested that most of the discrepancies between the anthropogenic $CO_2$ fluxes estimated by NGHGIs and global models reflect conceptual differences in how anthropogenic forest sinks and areas of managed land are defined.

The IPCC Guidelines for NGHGIs (IPCC, 2006; 2019a) distinguish three types of effects that can cause fluxes between land and the atmosphere: (1) direct human-induced effects, i.e. land-use changes and management practices; (2) indirect human-induced effects, i.e. human-induced environmental changes (e.g., changes in

atmospheric $CO_2$ concentration, nitrogen deposition, temperature, or precipitation) that affect growth, mortality, decomposition rates, and natural disturbances regimes; and (3) natural effects, including climate interannual variability and a background natural disturbance regime.

Global models - including BMs and IAMs (Integrated Assessment Models, used to estimate future emission pathways) - define managed forests as those that were subject to recent harvest and have not yet regrown to pre-harvest stock levels, whereas IPCC guidelines (IPCC 2006) and NGHGIs define managed forests more broadly as forests that fulfill social, economic and ecological functions, including protected areas or areas with fire prevention activities. In their larger managed forest areas, NGHGIs also generally consider most of the human-induced environmental changes as anthropogenic (see Methods), while the global model approach treats these changes as part of the non-anthropogenic, natural sink (estimated by DGVMs). Both approaches are valid in their specific contexts but are not directly comparable. Combining them to assess progress towards carbon neutrality or to quantify the remaining carbon budget might lead to biased results (Grassi et al., 2021). If the differences in land-use $CO_2$ fluxes between global models and NGHGIs are not reconciled or transparently explained, they may jeopardize the confidence in the mitigation potential of LULUCF, question fair burden-sharing of emissions reductions, and hamper an accurate assessment of the collective progress under the Paris Agreement's Global Stocktake. A greater effort of both communities for closer cooperation and mutual understanding is thus needed (Perugini et al., 2021).

- **Figure 1**-

A pragmatic approach has been proposed to address the differences between global models and NGHGIs. Specifically, Grassi et al. (2018) used a disaggregation of the natural land sink estimated by DGVMs to help "translate" the estimates from global models into results that are conceptually more comparable to NGHGIs. Grassi et al. (2021) refined this approach by filtering the DGVM results with a map of non-intact forests (taken as proxy of countries' managed forest) to reconcile the gap between NGHGIs and five IAMs for the period 2005-2015, and applied this approach to various emissions scenarios until 2100. This refined approach was also included in the Global Carbon Budget (table 9, Friedlingstein et al., 2022) to reconcile the difference in historical global land-use estimates between BMs and NGHGIs. The findings and recommendations from Grassi et al. 2021, i.e. that adjustments should be made whenever a comparison between LULUCF fluxes reported by countries and the global models is attempted, are reflected in the work of the IPCC Sixth Assessment Report (e.g., box 3 in Chapter 3 and box 6 in Chapter 7, IPCC, 2022) and in UNFCCC reports of high policy relevance, such as the Synthesis report of NDCs (UNFCCC, 2021) and the Synthesis report for the technical assessment component of the first Global Stocktake (UNFCCC, 2022b). In the absence of these adjustments, collective progress under the Global Stocktake, based on NGHGIs, would appear better than it is.

The present study illustrates and discusses in more detail the reconciliation between BMs and NGHGIs briefly shown in Friedlingstein et al. (2022) at global level, analysing the period 2000-2020. It includes slight updates

in the managed forest area and a greater level of disaggregation in terms of land categories (forest land, deforestation, organic soils, other land uses) by regions and countries. The study tests hypothesis that most of the previous large differences are reconciled globally as well as regionally and for large countries if land-use fluxes by BMs are made conceptually more comparable to NGHGIs. This is of major importance as the different definitions of the anthropogenic $CO_2$ sink (global models vs. NGHGIs) may have implications for a fair and realistic allocation of mitigation targets across countries (Schwingshackl et al., 2022).

While this study focuses on the reconciliation of the main conceptual difference between BMs and NGHGIs, it also discusses other methodological differences. Furthermore, we indicate priority areas for future research, discuss our results in the context of the Global Carbon Budget and outline a path toward a more robust operationalization of the comparison between global models and NGHGIs. The ultimate aim is increasing the confidence in land-use flux estimates reported by NGHGIs, which is key to foster investments in LULUCF mitigation and to assess the collective progress under the Paris Agreement's Global Stocktake.

**Methods**

**Global models**

Two fundamentally different types of global models are used to simulate the $CO_2$ exchange between the terrestrial biosphere and the atmosphere (Friedlingstein et al., 2022): bookkeeping models (BMs) and dynamic global vegetation models (DGVMs).

BMs track changes in the carbon stocks of areas undergoing land-use and land-cover change using predefined carbon stocks and rates of growth and decay for different types of vegetation and soil carbon. Here we use

results for 2000-2020 from the simulations of three BMs used in the Global Carbon Budget 2022 (Friedlingstein et al. 2022): BLUE (Hansis et al., 2015), OSCAR (Gasser et al., 2020), and H&N (Houghton and Nassikas, 2017). The net $CO_2$ flux from BMs (land-use change emissions, $E_{LUC}$, in Friedlingstein et al., 2022) includes $CO_2$ fluxes from deforestation, afforestation, harvest activity, shifting cultivation, regrowth of forests and other natural vegetation-types following wood harvest or abandonment of agriculture, and transitions between other

land types. This flux includes the direct human-induced effects only, as described by the IPCC Guidelines (2006, 2019a). Typically, BMs limit the maximum biomass of post-harvest regrowth up to the pre-harvest carbon stock levels. Emissions from peat burning and peat drainage are added from external datasets (see Appendix C2.1 in Friedlingstein et al., 2022). In terms of the land-use change data used to drive the models, BLUE is based on LUH2 (Hurtt et al., 2020), H&N is based on FAO data (FAOSTAT, 2021) and OSCAR uses both LUH2 and

FAO as input data. BMs generally do not include the $CO_2$ fluxes associated with natural disturbances.

DGVMs simulate ecosystem processes (primary productivity, autotrophic and heterotrophic respiration), their response to changing $CO_2$, climate, anthropogenic land-cover changes, and, depending on the model, additional processes such as management, nitrogen inputs and a limited range of natural disturbances, as well as natural vegetation dynamics in response to environmental changes (Sitch et al., 2015). In our study, we use results from

the DGVM model intercomparison project TRENDY. TRENDY (Sitch et al., 2015) is conducted regularly as part of the Global Carbon Budgets, where the DGVMs' estimates serve two purposes. First, the DGVMs provide an estimate of the 'natural terrestrial sink', conducting a historical simulation that accounts for environmental changes (climate, $CO_2$ concentration, nitrogen deposition) but not accounting for land-use changes (called "S2" in the TRENDY protocol, see Obermeier et al., 2021). This 'natural sink' includes both the indirect human-

induced and natural effects as described above (IPCC 2006; 2019a). Second, the DGVMs provide an estimate of the uncertainty of the land-use emissions estimated by BMs. For this second purpose a second similar simulation is conducted that accounts for land-use changes (called "S3"). These land-use estimates are used only as uncertainty assessment to the BMs' estimates instead of providing the land-use emissions term directly, because DGVMs differ greatly in terms of their completeness of land management processes (Friedlingstein et

al., 2022, Arneth et al., 2017).

By contrast, BMs typically rely on observation-based data, such as carbon densities of different biomes, and therefore, implicitly, have a more complete representation of land management processes than DGVMs. For example, primary and secondary ecosystems in BMs may have different carbon densities to reflect degradation, and the observation-based soil carbon estimates for cropland implicitly capture all land management processes such as tillage, fertilization and harvesting - processes that only some DGVMs have implemented. However, by not representing such land management activities in a process-based way, BMs represent average values (e.g. country- or biome-averages) rather than distinguishing different levels of intensification or specific forms of management, such as forest thinning.

Another reason for which the DGVMs' estimates of land-use emissions are not used directly in the Global Carbon Budget, in addition to the BMs', lies in the fact that their estimates are not directly comparable to BM estimates or any observable carbon fluxes because they include the 'loss of additional sink capacity', i.e., the difference between the actual land sink under land-cover changing in response to land-use changes and the counterfactual land sink under pre-industrial land-cover (Gasser and Ciais, 2013; Pongratz et al., 2014, Obermeier et al. 2021). Since we focus here on land-use fluxes estimated by BMs as used by the Global Carbon Budget and by the IPCC assessment reports, we do not further discuss the differences between DGVM and BM estimates of land-use fluxes (see Obermeier et al., 2021, for an in-depth discussion of this topic).

In this study, we use the DGVMs instead for their first purpose in the Global Carbon Budgets, i.e. for their estimate of the natural terrestrial $CO_2$ fluxes. We consider 16 DGVMs, derived from the same model run as used in Friedlingstein et al., 2022 (i.e., the S2 run from TRENDY v11, that uses transient $CO_2$ and climate forcing but keeps land cover and land use constant at pre-industrial values, and denoted as SLAND). The S2 simulations of DGVMs assume all forests to be natural and do not explicitly simulate secondary forests.

Estimates from both BMs and DGVMs include all carbon pools covered in the IPCC reporting guidelines for the NGHGIs (IPCC, 2006), although with different aggregation.

**National Greenhouse Gas Inventories (NGHGIs)**

The UNFCCC requires its Parties to report NGHGIs of anthropogenic GHG emissions and removals. At present, this obligation differs for Annex I Parties (AI, advanced economies with annual GHG reporting commitments under the UNFCCC) and non-Annex I Parties (NAI, countries with less stringent reporting commitments), but a more harmonised reporting under the Paris Agreement is expected through Biennial Transparency Reports starting by the end of 2024. While NGHGIs follow scientific methodological guidelines (e.g., IPCC, 2006; 2019a), they should not be primarily seen as scientific reports, but rather as a way to monitor the impact of mitigation actions in each country.

In this study, we use the most up-to-date and complete compilation of country-level LULUCF estimates (Grassi et al., 2022). This database builds on a detailed analysis of a range of country submissions to the UNFCCC and

is complemented by information on managed and unmanaged forest areas. Specifically, for AI countries, data are from annual GHG inventories (including a complete time series from 1990 to 2020). For NAI countries, the most recent and complete information was compiled from different sources, including National Communications (NC), Biennial Update Reports (BUR), submissions to the framework REDD+ (Reducing Emissions from Deforestation and Forest Degradation), and NDCs. The data are disaggregated into fluxes from forest land, deforestation, organic soils, and other sources (Table 1), to facilitate comparison with BMs. This database includes LULUCF data from 185 countries, covering the vast majority of the global forest area (i.e., the land-use category for which countries report most of the emissions and removals). To ensure a complete time series from 2000 to 2020, which is often not yet available in NAI countries, gaps were filled using standard statistical methods, with the aim to maintain the levels and trends of the underlying, reported raw data. The overall gap-filling rate is 48 % (0 % for AI, 62 % for NAI countries); however, when normalised by the contribution to the global carbon flux values, the gap-filling rate becomes 30% of the absolute total flux (0 % for AI, 40 % for NAI countries), indicating that most of the NAI countries where the biggest fluxes occur report relatively complete time series. Compared to Grassi et al. (2022), here we updated the values for the Democratic Republic of Congo (DRC), that in its BUR1 introduces a time series for the forest sink.

In terms of reported carbon pools, the situation varies depending on the country and the land category. The IPCC guidelines (2006; 2019) distinguish living biomass (above- and below-ground), dead organic matter (dead wood and litter), soils (mineral and organic) and harvested wood products (sometimes referred to as a separate category rather than a carbon pool). The vast majority of AI countries report the $CO_2$ fluxes from the carbon pools in case of land-use changes (e.g., forest converted to settlements, cropland converted to forest, grassland converted to cropland), and from the most important carbon pools in case of land uses that remain unchanged (e.g., biomass in forest land remaining forest land, soil in cropland remaining cropland). The NAI countries typically report the $CO_2$ fluxes from living biomass on deforestation and, in the vast majority of cases, on forest land. For the other pools the situation is less clear. Dead organic matter, mineral soils and harvested wood products are reported by the largest NAI countries (including Brazil, China, India, Indonesia, Mexico) but are often not considered by other NAI countries. $CO_2$ fluxes from organic soils are reported only by a few NAI countries (e.g., Indonesia). Overall, the quality and quantity of the LULUCF data submitted by countries to the UNFCCC significantly improved in recent years, but important gaps and areas of improvement still remain. For example, most NAI countries still do not explicitly separate managed vs. unmanaged forest land, a few report implausibly high forest sinks or inconsistent estimates among different reports, and several report incomplete estimates (especially in Africa, where many countries still have low national capacity for reporting). Yet, these gaps are expected to be progressively filled under the Paris Agreement Transparency Framework reporting (from the end of 2024 onwards). For more details, including a discussion of the differences between the NGHGI database, the UNFCCC GHG data interface (UNFCCC, 2022b), and the FAOSTAT Land-use emission database (Tubiello et al., 2021), see Grassi et al. (2022).

Due to the impossibility of providing widely applicable methods to disentangle direct and indirect human-induced effects and natural effects on land GHG fluxes through direct observations (e.g., national forest inventories), the IPCC Guidelines adopted the 'managed land' proxy (IPCC, 2006; 2010; 2019a) as a pragmatic approach to facilitate NGHGI reporting. Accordingly, anthropogenic land GHG fluxes are defined as all those occurring on managed land, i.e. where human interventions and practices have been applied to perform production, ecological, or social functions. GHG fluxes from unmanaged land are not reported because they are assumed to be non-anthropogenic. The specific land processes included in NGHGIs depend on the estimation method used, which differs in approach and complexity among countries. Grassi et al. (2018) concluded that most countries report both direct and most of the indirect human-induced effects on managed forests. Indirect effects are included especially when the stock-difference approach or recent forest growth factors are used to estimate net emissions and removals. With regard to natural effects (including interannual climate variability and natural disturbances), these are included in different ways by different NGHGIs. Since most NGHGIs rely on periodic measurements, interannual climate variability is seldom explicitly represented. By contrast, natural disturbances, such as fires, insects, and wind throws, are included in most NGHGIs with the exception of Canada and Australia. Following the IPCC guidelines (IPCC 2019a), these two countries implement a 'second-order approximation' for anthropogenic $CO_2$ fluxes (in principle, a refinement of the managed land proxy) and disaggregate the GHG emissions and subsequent $CO_2$ removals that are considered to result from natural disturbances within their NGHGIs. These fluxes are reported separately in the NGHGI: the average net emissions that were disaggregated for the period 2000-2020 amounted to about 104 $MtCO_{2eq}$ yr$^{-1}$ in Canada (Canada, 2022) and 39 $MtCO_2$ yr$^{-1}$ in Australia (Australia, 2022).

While the different approaches to include direct and indirect human-induced effects represent the main conceptual difference between NGHGIs and global models, other differences exist as well. For example, differently from BMs, the IPCC methodological guidance does not assume that post-harvest forest regrowth is limited up to the pre-harvest carbon stock levels. In NGHGIs, these levels might be exceeded not only due to the impact of indirect human-induced effects but potentially also due to improvements in management practices that stimulate higher productivity. These improvements may lead to greater biomass density due to direct effects not explicitly simulated by BMs (e.g., Erb et al., 2013; Kauppi et al., 2020), such as greater site fertility (due to discontinued litter raking), selection of trees with higher growth rates or stocking density, or a better-regulated competition among trees (due to thinning).

- Table 1-

The majority of countries use the net land $CO_2$ flux reported in NGHGIs to assess compliance with their NDCs and track progress of their long-term (i.e., 2050) emission reduction strategies under the Paris Agreement (Grassi et al., 2017). However, some countries expressed the intention to apply specific accounting rules to these estimates, which may contribute to the lack of clarity in the land contribution towards the NDCs (Fyson and

Jeffery, 2019). These accounting rules aim to better quantify the impact of additional mitigation actions on the net land $CO_2$ flux by, for example, discounting the impact of natural disturbances and forest age-related dynamics (Kurz et al., 2018; Grassi et al., 2018; IPCC 2019a). Since we focus here on the estimated $CO_2$ fluxes, we consider the estimates reported in NGHGIs in their managed land irrespective of their potential future filtering through accounting rules.

**Approach to harmonise land-use flux estimates of global models and national GHG inventories**

To harmonise the land-use fluxes of BMs and NGHGIs, we apply an approach similar to the one described by Friedlingstein et al. (2022), with minor updates and a much greater disaggregation of the results to allow a more detailed comparison with NGHGIs.

Building on previous studies (Grassi et al., 2018; 2021), we added the 'natural' $CO_2$ sink estimated by 16 DGVMs ($S_{LAND}$, Friedlingstein et al. 2022, including both indirect human-induced and natural effects as defined by IPCC, 2006) in countries' managed forest areas to the direct anthropogenic land-use flux estimates from BMs ($E_{LUC}$) for the period 2000-2020. To determine $S_{LAND}$ in managed forests, the following steps were taken: Spatially gridded data of "natural" forest Net Biome Production (NBP = $S_{LAND}$) were obtained from the TRENDY v11 S2 runs performed for the Global Carbon Budget 2022 (Friedlingstein et al., 2022) for the period 2000-2020. From the same data, we also disaggregated the sink from forest soils, computed as the difference between the annual mean stock of two consecutive years (available only for nine DGVMs). It should be noted that, although the differences in carbon uptake by natural and secondary forests are not considered by these runs, the DGVMs gridded results capture the spatial heterogeneity of the sink in terms of different forest types, soil types, and local climate.

The results were then filtered by the area of managed forest, according to a protocol developed for this study (see online repository, Grassi et al., 2023). Essentially, data were first masked with the Hansen forest map for 2012 (Hansen et al., 2013), using a 20% tree cover threshold and following the FAO definition of forests (i.e., isolated pixels with maximum connectivity less than 0.5 ha are excluded). This ensures that the current forest area, and not the pre-industrial one used by S2 runs, is applied in this study. Results were then further masked by a map of "intact" forest for the year 2013 (Potapov et al., 2017). Intact forest is defined as areas without detected signs of human activity via remote sensing, which a previous study (Grassi et al., 2021) found to be a relatively good proxy for "unmanaged" forests in country reports. This way, we obtained $S_{LAND}$ separately for "intact" and "non-intact" forest areas, which we used as proxy for "unmanaged" and "managed" forest areas in the NGHGIs. Exceptions to the protocol above - which are expected to improve the comparability between BMs and NGHGIs - occurred for the three countries where the area of unmanaged land is most relevant (Canada, Brazil and Russia, accounting for 60% of unmanaged forest area in our study). For Canada and Brazil, this study uses the national gridded map used in the respective NGHGIs (Canada, 2022; Brazil, 2020), see Supplementary

Figure 1. For Russia, a country-specific adjustment to the Hansen forest map was implemented, allowing to obtain a better match with NGHGI data on managed forest and its regional distribution (Russian Federation, 2022), see Supplementary Figure 2.

At the global level, NGHGIs indicate about 3.7 and 0.7 billion ha of managed and unmanaged forest, respectively. In comparison, the IPCC Special Report on Climate Change and Land (IPCC, 2019b) indicates 2.9 and 0.9 billion ha of "forest managed for timber and other uses" and "forest with minimal human use", respectively. In terms of global ice-free land surface (ca. 13 billion ha), about 75-80% of land is considered to be under some form of human management (Erb et al., 2017; IPCC, 2019b), with the rest being unmanaged forested and unforested ecosystems (ca. 2 billion ha) or other land (barren, rock). By contrast, the BMs consider a much smaller area of managed forest than NGHGIs (e.g., 1.4 and 1.3 billion ha by BLUE and H&N, respectively). Finally, the areas used in this study - based on the combination of non-intact and intact forest plus country-specific information (for Russia, Canada and Brazil) - are about 3.3 and 0.8 billion ha for managed and unmanaged forest, respectively (Figure 2b, Supplementary Table 1). Australia is the country with the greatest difference between the area of managed forest used in this study (0.04 billion ha) and the NGHGI (0.13 billion ha, although the NGHGI assumes a large part of this area to be in carbon equilibrium).

It is important to note that we consider the impact of indirect effects (human-induced environmental change) only in forest areas, because most of the indirect land sink is expected to occur in forests. Non-forest land is scarcely reported in the NGHGIs of NAI countries, and at present no reliable proxy for managed land exists for non-forest land uses.

In order to facilitate the comparison of BMs' results with NGHGIs for specific regions and countries, our analysis provides detailed information both in terms of disaggregation of estimates for specific land categories (Table 1) and trends, which go substantially beyond the details of Friedlingstein et al. (2022).

**-Figure 2-**

Some methodological aspects should further be noted. First, only 5 of the 16 DGVMs used here (CABLE-POP, CLASSIC, JSBACH, OCN, and YIBs) indicated forest NBP at grid-cell level. For the other 11 DGVMs, all the NBP in a grid cell was allocated to forest in case a grid cell had forest. The average $CO_2$ flux in non-intact forest from the former 5 DGVMs was not significantly different ($P > 0.05$) from the other 11 DGVMs.

Second, using intact/non-intact maps for the year 2013 may lead to over- or underestimations of the managed forest area before or after 2013 (i.e. there is no trend in the intact forest area we used). However, since the net loss of total forest area in the period of our study (2000-2020) is very small compared to the total forest area in 2013 (around 2%), we can reasonably assume that the impact of our approach on the possible under- or over-estimation of the managed forest area (and the corresponding $S_{LAND}$) is minor: it could be roughly assumed that

our 2000 $S_{LAND}$ sink is underestimated by about 1% and the one in 2020 is overestimated by about 1% – well below the uncertainty from DGVMs.

Third, it may at first seem inconsistent to compare the NGHGI to the S2 simulation from DGVMs, as the S2 simulation provides estimates of the natural sink on the pre-industrial land-cover distribution, which globally had a much larger forest extent than present-day NGHGI. However, since we only use the natural carbon fluxes that occur on forest currently managed, our analysis excludes areas where forest cover has been lost historically due to agricultural expansion. Therefore, our harmonisation of estimates from NGHGIs and BMs is not confounded by this effect. The loss of additional sink capacity is implicitly reflected, however, in the rather large estimate of the non-forest natural sink in DGVMs shown in Fig. 10 below.

The data from this study are openly available in the online repository (Grassi et al., 2023) including, for each country, the land $CO_2$ flux from global models (for each BMs and the ensemble mean of the DGVMs) and from NGHGIs for each land-use category. Furthermore, in the same repository, we made available the detailed protocol to process the DGVMs results and the map of managed/non-intact forest used in this study.

## Results and discussion

### Quantifying the gap

While the difference in anthropogenic land-use $CO_2$ fluxes between global models and NGHGIs was discussed in other recent studies (Grassi et al., 2018; Grassi et al., 2021; Friedlingstein et al. 2022; Schwingshackl et al., 2022), the results presented here represent both an update based on data from the NGHGIs submitted in 2022, as well as an unprecedented level of disaggregation in terms of land categories (forest land, deforestation, organic soils, other), trends, regions, and countries. This disaggregation enables us to attribute the remaining

gap more precisely to different approaches and categories.

When the average net land-use $CO_2$ flux for 2000-2020 of the three BMs (4.8 $GtCO_2$ $yr^{-1}$) is compared to NGHGIs (-1.9 $GtCO_2$ $yr^{-1}$), a gap of about 6.7 $GtCO_2$ $yr^{-1}$ emerges from our study (Figure 3a). Most of the difference between BMs and NGHGIs is found for fluxes on forest land (3.5 $GtCO_2$ $yr^{-1}$, Fig. 3b), but discrepancies emerge also for deforestation fluxes (2.4 $GtCO_2$ $yr^{-1}$, Fig. 3c) and other fluxes (1.0 $GtCO_2$ $yr^{-1}$,

Fig. 3e). By contrast, the difference for fluxes from organic soils, which are added to the BM estimates from external datasets (see Table 1), is small (0.2 $GtCO_2$ $yr^{-1}$, Fig. 3d). In general, trends of BMs and NGHGIs are quite similar.

**-Figure 3-**

The gap in LULUCF fluxes between BMs and NGHGIs identified here (6.7 GtCO2 yr-1) is a bit higher than the gap of 5.5 $GtCO_2$ $yr^{-1}$ previously found between Integrated Assessment Models (IAMs, whose approach to estimating the anthropogenic land-use $CO_2$ flux is similar to BMs) and NGHGIs (Grassi et al., 2021). The difference reflects the updates made by both BMs and NGHGIs (Figure 4), which shifted the values downwards, i.e. smaller net emissions in BMs (due to updates in the land-use dataset used as input for BM simulations, see

Friedlingstein et al., 2022), and a greater net sink in NGHGIs (due to more complete reporting by NAI countries, Grassi et al., 2022).

**-Figure 4-**

**Bridging the gap between global models and national GHG inventories**

We split the forest sink that DGVMs attribute to the natural response of land to human-induced environmental change ($S_{LAND}$) into the parts occurring in non-intact (managed) and intact (non-managed) forests (Figure 5a). In the absence of country maps of managed forests (which we could only obtain for Canada and Brazil, plus we use country-specific information from Russia), we use the intact/non-intact forest map as a proxy for unmanaged/managed forests (see Methods, and Supplementary Table 1). Four-fifths of the global forest sink (-8.1 $GtCO_2$ $yr^{-1}$ for the period 2000-2020) occur in non-intact (managed) areas, which similarly represent about four-fifths of the ~4 Billion ha of world's forests (Figure 2b).

In this study, we use the average $S_{LAND}$ in non-intact forest from 16 DGVMs models (Figure 5b) as a proxy for the sink from indirect human-induced effects in managed forest (see Figure 1), which is assumed to be included in the majority of NGHGIs. While exceptions to this assumption exist - e.g. the methods used by Australia and Canada imply that only a part of these indirect effects is included in their NGHGIs (see Methods) - the available information indicates that the majority of countries report most of the indirect human-induced effects on their managed land (Grassi et al., 2018).

The sink in non-intact forest estimated by Grassi et al. (2021), using one DGVM only, was equal to -5.0 $GtCO_2$ $yr^{-1}$ for the period 2005-2020 (Supplementary Table 8, Grassi et al., 2021). In this study, for the period 2000-2020, 16 DGVMs estimate a sink in non-intact forest of -6.4 $GtCO_2$ $yr^{-1}$. In both cases, the adjustments based on these sink estimates reconcile most of the gaps identified between the anthropogenic land-use $CO_2$ flux estimated by NGHGIs and by global models (either IAMs or BMs). The remaining gaps – i.e., about -0.5 $GtCO_2$ $yr^{-1}$ in Grassi et al. (2021) for 2005-2020, and -0.3 $GtCO_2$ $yr^{-1}$ in this study for 2000-2020 – are well within the uncertainty of the respective datasets (see e.g. the large variability of DGVM estimates in Figure 5b).

**-Figure 5-**

If the sink from non-intact forest is added to the original results from BMs, the adjusted results for LULUCF agree much better with the sum of NGHGIs (Figure 6a). At the same time, this adjustment leads to a forest sink that is higher for the adjusted BMs compared to NGHGIs (Figure 6b). Overall, these findings suggest that the conceptual difference in defining the anthropogenic forest sink (BMs versus NGHGIs) explains most of the gap but not all. To gain more insights into the remaining differences, we compare below the results from BMs and NGHGIs in the following for individual land-use categories at global, regional, and country-level.

**-Figure 6-**

For LULUCF, the match between BMs and NGHGIs improves considerably after the adjustments both at the global level – where the original gap is reduced from 6.7 $GtCO_2$ $yr^{-1}$ to 0.3 $GtCO_2$ $yr^{-1}$ – and for AI and NAI countries (Figure 7a). While the same pattern is confirmed for most of the regions and countries analyzed (Figures 7b and 7c), in some cases, the adjustment does not reduce the gap (Canada, DRC), as also noted by

Schwingshackl et al. (2022). Furthermore, after the adjustment, a large discrepancy remains in Asia, with BMs + DGVMs estimating higher net emissions than NGHGIs (including in China and India). At the global level, this discrepancy is partly compensated by differences in the opposite direction in Africa, where BMs + DGVMs estimate lower net emissions than NGHGIs.

**-Figure 7-**

We further disaggregate the LULUCF fluxes into four main land categories, separately for AI and NAI countries (Figure 8). The forest sink in the adjusted BMs is higher than in NGHGIs, especially for NAI countries. According to Grassi et al. (2022) while some NAI countries report implausibly high estimates of the forest sink

(Central African Republic, Mali, Namibia, Malaysia, and Philippines), others do not report any estimate of the sink in forest land (e.g. Tanzania, Mozambique, Guyana). In addition, not all NGHGIs include all recent indirect human-induced effects (e.g., due to $CO_2$ fertilization), and thus may underestimate the forest sink relative to the adjusted estimates from BMs.

For deforestation, organic soils, and other fluxes, the match between BMs and NGHGIs is reasonable for AI

countries (gap not higher than 0.1 $GtCO_2$ $yr^{-1}$ for each category), while a larger gap is found for NAI countries (2.4 $GtCO_2$ $yr^{-1}$ for deforestation and 0.9 $GtCO_2$ $yr^{-1}$ for other). The greater differences for NAI countries may be due to a far less complete reporting in NGHGIs compared to AI countries (see Methods).

While the separation of BMs' results into various land categories helps the comparison with the NGHGIs, an important source of uncertainty (especially for NAI countries) is how the fluxes from shifting agriculture are

allocated, i.e., if they are placed into forest, deforestation, or other. Specifically, in this study BLUE and OSCAR allocate emissions from shifting agriculture under "deforestation" and any subsequent removals under "forest" (e.g. for OSCAR, this corresponds to +3.5 $GtCO_2$ $yr^{-1}$ under deforestation and -2.5 $GtCO_2$ $yr^{-1}$ under forest for the period 2000-2020); H&N allocates emissions from shifting agriculture under "deforestation" only after the first conversion occurs (this corresponds to about +1.1 $GtCO_2$ $yr^{-1}$ for the period 2000-2020), and thereafter the

emissions and removals (overall a small net flux) are allocated to "other fluxes". The quantitative importance of shifting agriculture for $CO_2$ fluxes is also confirmed by Harris et al. (2021). For NGHGIs, it is often unclear if and under which categories the fluxes due to shifting agriculture are reported. While the above difference may help to explain the larger emissions from deforestation in BMs than in NGHGIs, and also the larger forest sink in the adjusted BMs (i.e. including the natural sink estimated by DGVMs for the managed forest area) than in

NGHGIs, the lack of reliable information from most NAI represents one of the biggest sources of uncertainty in our comparison at the level of individual land-use categories.

A more detailed disaggregation of the results by region and large countries is illustrated in Figure 9, and more complete results for the main countries are provided in Supplementary Table 1. Results for all countries, including the yearly values of $CO_2$ fluxes from BMs, DGVMs and NGHGIs, are available in the online

repository (Grassi et al., 2023).

-Figure 8-

-Figure 9-

For forest land, the inclusion of natural fluxes in non-intact forests improves the match between BMs and NGHGIs in AI countries but not in NAI countries, where the adjusted estimates from BMs' result in a substantially higher sink than NGHGIs (Figure 9a). This may be, at least in part, explained by the possible different treatment of shifting cultivation by BMs and NGHGIs (see above). Furthermore, even where the regional match for forest land is good, some discrepancies remain at the country level (Figure 9b). In Canada, for example, while the original BMs' result is close to the NGHGI, the adjusted one yields a much larger sink (by 0.36 $GtCO_2$ $yr^{-1}$). This may be explained at least in part by the fact that Canada uses empirical growth and yield curves. While environmental change (e.g., climate and $CO_2$ forcing) could enhance tree growth over time (but see Girardin et al., 2016), empirical yield curves represent average growth rates measured over decades, therefore not fully including the recent impact of indirect human-induced effects – an approach that is conceptually similar to the original BMs' results. In Russia, while the net sink in the original BMs' results is much lower than the NGHGI, the adjusted BMs' results yield a larger sink (by 0.43 $GtCO_2$ yr-1) than the NGHGI. The latter difference may be explained by a possible underestimation of the sink in the NGHGI, as noted by recent studies (Schepaschenko et al. 2021). In Asia, the main discrepancies after the adjustment are observed in Indonesia and Myanmar, where the adjusted forest sink of BMs (in particular BLUE and OSCAR) is greater than the NGHGI sink. On the other hand, for China, the gap between the original BMs estimate and the NGHGI reduces considerably after the adjustment. In Africa, the greater sink in the adjusted BMs' results is mainly explained by the large discrepancy (0.36 $GtCO_2$ yr-1) observed for DRC (see Supplementary Table 1). In Latin America, the larger forest sink by BMs + DGVMs compared to the NGHGIs is mainly due to differences found in Brazil (0.22 $GtCO_2$ $yr^{-1}$), Colombia (0.22 $GtCO_2$ $yr^{-1}$) and Bolivia (0.12 $GtCO_2$ $yr^{-1}$. Overall, the natural forest sink estimated by DGVMs compares quite well with the net sink estimated from ground plots of intact old-growth tropical forests (Hubau et al. 2020). Specifically, Hubau et al. (2020) estimated a net sink of about -2.2 $tCO_2$ $ha^{-1}$ $yr^{-1}$ in Africa (2000-2015) and -1.5 $tCO_2$ $ha^{-1}$ $yr^{-1}$ in the Amazon (in 2000-2011), while DGVMs estimated a net sink of -2.4 $tCO_2$ $ha^{-1}$ $yr^{-1}$ in Africa and -1.8 $tCO_2$ $ha^{-1}$ $yr^{-1}$ in the Amazon in the same period (although with large variability among models, see Figure 5b). The potential slight overestimation of the sinks in Africa and the Amazon by DGVMs could explain part of the remaining difference between adjusted BMs' results and NGHGIs.

For deforestation, regional results from BMs broadly agree with NGHGIs except in Asia and Africa (Figure 9c). In Asia, BMs estimate higher emissions in Indonesia (by 0.65 $GtCO_2$ $yr^{-1}$), China (by 0.22 $GtCO_2$ $yr^{-1}$), Myanmar (by 0.20 $GtCO_2$ $yr^{-1}$), India (by 0.13 $GtCO_2$ $yr^{-1}$), and Vietnam (by 0.11 $GtCO_2$ $yr^{-1}$). By contrast, in Africa, BMs estimate smaller emissions than NGHGIs in Nigeria (by 0.21 $GtCO_2$ $yr^{-1}$) and Central African Republic (by 0.08

GtCO$_2$ yr$^{-1}$) (Supplementary Table 1). In Latin America, BMs estimate higher emissions than NGHGI in Brazil (by 0.21 GtCO$_2$ yr$^{-1}$). Estimates of deforestation fluxes from models strongly depend on the underlying datasets, including the spatial resolution of the land cover data or statistics used (Winkler et al., 2021). For example, a recent study (Ganzenmüller et al., 2022) concluded that deforestation emissions based on high-resolution activity data substantially lower the previously estimated emissions using the LUH2 datasets (used here by BLUE and OSCAR). On the other hand, some NGHGIs do not report emissions from gross deforestation (for example, China and India).

It is worth noting that, for forest land and deforestation, a much better match between the adjusted BMs and NGHGIs is obtained if these two categories are combined, probably because the separation of CO$_2$ fluxes between these categories is done differently by BMs and NGHGIs (e.g. see comment above on shifting cultivation).

For organic soils, results from the external datasets that are added to BMs broadly agree with NGHGIs. In Asia, the biggest differences are found in Indonesia, where the NGHGI reports higher emission than BMs, and in China, where the NGHGI does not report this category separately.

In the category 'other' (cropland, grassland, wetlands, settlements; i.e. land uses that are more poorly included in NGHGIs in general), the large difference in Asia is mainly due to a large sink in agricultural lands reported by India and China, whereas BMs report a source for these countries. While this may be partly due to the fact that BMs estimate only land-use changes for agricultural lands (e.g., grassland conversion to cropland and not 'cropland remaining cropland'), the large sinks reported by India (for cropland) and China (for cropland, grassland, and wetlands) are not well documented. BMs report greater emissions than NGHGIs in Africa and Latin America, presumably mainly due to the incomplete reporting of these categories in NGHGIs. For the USA, the greater sink from the NGHGI compared to BMs is partly explained by the large sink reported in settlements (-0.13 GtCO$_2$ yr$^{-1}$, mainly due to urban trees), a category not estimated by BMs.

With regard to the allocation of fluxes to the various carbon pools, the comparison between global models and NGHGIs is hampered by different definitions of carbon pools and by incomplete estimations by NGHGIs (especially for NAI countries). Nevertheless, based on the available data, mineral soils do not seem to represent a major source of difference in land use CO$_2$ fluxes between global models and NGHGIs, at least in forest land. According to NGHGI data from AI countries for the category forest land, the vast majority of the forest sink is reported in the living biomass, with mineral soil and dead organic matter representing, respectively, 7% and 11% of the total net sink for the period 2000-2020 (excluding organic soils). This information is broadly in line with the results from global models. Data from the TRENDY v11 dataset (for nine DGVMs only), for example, show that the sink in forest soils represents about 10% of the overall forest sink in AI countries during the same period. For all land uses, the BMs' results indicate a net source from mineral soils (about 1.5 GtCO$_2$ yr$^{-1}$ from BLUE and 0.6 GtCO$_2$ yr$^{-1}$ from H&N for 2000-2020), with emissions associated with land-use changes (mostly deforestation) and a small sink in forest land. Overall, we argue that a more comprehensive analysis of fluxes

in different carbon pools should be prioritized in future studies comparing global models and NGHGIs, along with analyses of possible lateral fluxes that might be overlooked by both BM and NGHGIs.

**Our results in comparison to other global studies**

Figure 10 summarizes our results in comparison with the main components of the Global Carbon Budget 2022 and with other recent literature. For the Global Carbon Budget 2022 (Friedlingstein et al., 2022), a net land-to-atmosphere sink of -5.9 $GtCO_2$ $yr^{-1}$ is obtained (Figure 10c) when the direct anthropogenic flux from BMs (4.8 $GtCO_2$ $yr^{-1}$ for the period 2000-2020, Figure 10a) is added to the natural terrestrial flux from DGVMs (-10.8

$GtCO_2$ $yr^{-1}$, Figure 10b). The natural terrestrial sink closely matches the land sink (-10.9 $GtCO_2$ $yr^{-1}$) estimated as the residual from the other flux components of the global carbon budget (i.e., fossil fuels, atmosphere, and ocean - which are less uncertain than the natural terrestrial sink - and land-use change, Friedlingstein et al. 2022). This consistency arising from using two independent approaches provides confidence on the estimated size of the net global land sink.

**-Figure 10-**

Consistent with Friedlingstein et al. (2022), the adjusted BMs' results obtained in our study (Figure 10d, i.e. Figure 10a plus the striped managed forest area in Figure 10b) compare well with the NGHGIs (Figure 10e), with both datasets indicating a relatively small net sink in managed land globally. This sink results from a large

net sink in temperate and boreal regions (mostly represented by AI countries) and a small net source in the tropics (mostly represented by NAI countries, Figure 7a). However, a few lines of reasoning and evidence suggest a possible underestimation of the net global sink in managed land, in both the adjusted BMs' results and the NGHGIs.

First, the fact that about 75-80% of land is under some form of management (Erb et al., 2017; IPCC, 2019)

could suggest, as a first approximation, that a similar share of the total natural terrestrial sink due to indirect effects (Figure 10b) - i.e. about -8.0 to -8.5 $GtCO_2$ $yr^{-1}$ -, is in managed land. If this hypothetical sink is summed up to the original BMs' results (about 4.8 $GtCO_2$ $yr^{-1}$, Figure 10a), it would result in a net global sink in managed land close to -3.5 $GtCO_2$ $yr^{-1}$. From another perspective: assuming the net land-atmosphere flux from the Global Carbon Budget (Figure 10c) as a valid estimate, it is questionable that the ca. -4 $GtCO_2$ $yr^{-1}$ sink difference with

our estimates for managed land (i.e., Figures 10d and 10e) occurs in the relatively small land area that has remained unmanaged.

Second, the net global sink in our estimates (both adjusted BMs' results and NGHGIs) is lower than most of the recent literature. For instance, Deng et al. (2022) – an inverse modelling study corrected for $CO_2$ emissions induced by lateral fluxes to produce terrestrial carbon stock changes estimates that can be compared to our study

– estimated a sink of -5.1 $GtCO_2$ $yr^{-1}$ in all land for the period 2007-2017 (Figure 10f), mostly from managed lands (identified as non-intact forest, similarly to our study). Deng et al. (2022) also indicated larger sinks over managed lands than the NGHGIs in Russia, EU and Canada, suggesting that some carbon storage processes may be underestimated in NGHGIs, such as the carbon increase in trees outside forests (urban green areas, trees on grassland and cropland, arctic shrubs) and in soils. By integrating remote sensing data with a map of biomass complemented by forest growth curves, harvest, and fires removals, Harris et al. (2021) estimated a net global forest sink of -7.6 $GtCO_2$ $yr^{-1}$ for the period 2000-2019 (including emissions from deforestation), mostly occurring on non-primary (or non-intact) forests of temperate and boreal regions. For the same period, Xu et al. (2021) estimated a net global sink of -3.2 $GtCO_2$ $yr^{-1}$ for aboveground biomass only (including intact forest), based on annual biomass maps obtained with optical and LiDAR data and a machine learning model. Similarly, Yang et al. (submitted) inferred a net global sink in above-ground biomass of -1.9 $GtCO_2$ $yr^{-1}$, mostly in extra-tropical regions, using a global L-Band vegetation optical depth (data from https://carbonstocks.kayrros.com). It should be noted that the latter two studies do not include belowground biomass and non-biomass carbon pools (such as soils), and thus likely underestimate the global net sink.

Third, the sink due to both direct effects (estimated by BMs) and indirect effects in managed forest (i.e. our adjustment to BMs, estimated by DGVMs) might be underestimated. BMs have a relatively simple representation of the management of forests and other land uses, e.g. they include harvest but not other practices that typically stimulate higher forest productivity (e.g., forest thinnings) and would thus cause larger sinks (e.g., Kauppi et al., 2020). The DGVM simulation used here (i.e., S2, including only indirect and natural effects but no land-use change) does not include a mechanistic description of forest management (i.e., magnitude and frequency of harvesting operation, stocking density), and forest demography (age-class structure, Pugh et al. 2019) in general, and therefore cannot predict the impact of changes in management and age dynamics on the intensity of indirect effects. As a result, our adjustment method assumes that the sink per unit area due to indirect effects is identical in managed and unmanaged forests (or in young and old forests) under the same climate conditions. Although rising $CO_2$ stimulates photosynthesis, the overall impact on the net carbon sink is complex (taking resource limitations, respiratory losses, and other factors into account) and is an active area of research (Walker et al., 2021). There is some evidence suggesting that the effect of rising $CO_2$ on the net sink could be larger in managed or young than in pristine or mature ecosystems (Walker et al. 2019, Jiang et al., 2020, Gundersen et al., 2021). Given the limitations of the DGVM ensemble in modelling forest successional stages, this reasoning implies that the sink in managed forests could be larger than the model ensemble estimates (in Figure 10b the dashed green area should be bigger and the dark green area smaller).

Lastly, there are a number of reasons why NGHGIs are uncertain, and possibly underestimate the global net sink in managed lands, including:

(i) While a few NGHGIs possibly underestimate emissions or overestimate removals of $CO_2$, these effects are likely counterbalanced by the incomplete reporting in terms of land uses (especially for non-forest land in

developing countries, Grassi et al., 2022) and carbon pools (especially for soil). The incompleteness of estimates increases the uncertainties;

(ii) NGHGIs do not always fully include the impact of human-induced environmental change when old data are used (e.g., Schepaschenko et al., 2021), and this may cause an underestimation of the sink estimate;

(iii) NGHGIs do not always include all the fluxes occurring on managed land, if no methodology exists for specific fluxes under the IPCC guidelines (2006) or if these fluxes are considered as non-anthropogenic. A relevant example is undrained wetlands in Russia: here, peatlands represent more than 20% of the country (including shallow peat deposits), 38% of this area includes woody vegetation (Vompersky et al., 2011), and part of it falls under managed forests. Due to the slow decomposition of organic matter under conditions of

oxygen deficiency, the carbon reserves in the peat deposit are constantly increasing (Joosten et al., 2016). A recent study (Korotkov et al., 2018) estimates that wetland ecosystems in Russia are a major carbon sink (about -0.86 $GtCO_2$ $yr^{-1}$) and simultaneously a source of methane emissions. These fluxes are not included in the NGHGI, unlike drained rewetted organic soils.

**Way forward: how to operationalize comparisons between global models and NGHGIs?**

This study focuses on the main conceptual difference between BMs and NGHGIs. By harmonizing the way the anthropogenic land use fluxes are estimated, our approach reconciles most of the current large gap in land use $CO_2$ fluxes between the two datasets, and provides a greater level of spatial and process details than previous

analyses.

We acknowledge the open debate on how to reconcile global models and NGHGIs, e.g. see the instructive discussion between M. Meinshasuen and S. Federici on adjusting models vs. adjusting country data (https://doi.org/10.5194/essd-2022-245-CC1). A pragmatic interim solution that we propose is to adjust the global models' results if the analysis is partly or predominantly focused on country or regional levels, and

considering adjusting the sum of country data to the models' results if the analysis is focused on climate mitigation efforts at the global level relative to modeled emissions pathways. This approach, followed also by the UNEP Emissions Gap Report 2022 (UNEP 2022, see Box 2.1 therein) ensures that country estimates are consistent with those reported by countries themselves to the UNFCCC, and that global estimates are consistent with the carbon cycle, scenarios and climate science literature used in the IPCC Assessment Reports. Given the

focus of this study on regional and country-level estimates and on disaggregating fluxes into different categories, we here adjusted BMs' results to country data.

Irrespective of the direction of the adjustment, the suggested approach for harmonizing land-use flux estimates of BMs and NGHGIs should be seen as a short-term and pragmatic fix based on existing data, rather than a definitive solution to bridge the difference between global models and NGHGIs. Additional steps are needed -

from both global models and NGHGIs - to understand and reconcile the remaining differences, some of which are relevant at the country level, and to operationalize future comparisons between global models and NGHGIs.

For global models, other studies have already highlighted many fundamental challenges (Pongratz et al. 2021), including the need for a better representation of land management processes and forest demography. Here we highlight the advantage of providing more disaggregated results for the BM estimates to increase comparability with NGHGIs, relative to aggregating data only into gross sources and gross sinks. This is because, for example, gross sources include both deforestation and forest harvest, while gross sinks include forest regrowth after harvest and afforestation. By contrast, NGHGIs aggregate fluxes from forest harvest and forest regrowth under the category 'forest land', with afforestation reported in a distinct subcategory while deforestation is reported in non-forest land uses (i.e., under the final land use category). In this regard, the land-use categories used here and by the Global Carbon Budget 2022 may offer a blueprint for future studies because they represent a minimum common denominator between the information provided by most NGHGIs - being aware of the large differences in the quality of reporting between AI and NAI countries - and the typical outputs produced by BMs.

NGHGIs could be made more comparable to global models if they either restrict their estimates to direct anthropogenic fluxes only (like BMs) or if they broaden their scope to the entire national territory (without distinction between managed and unmanaged lands). The first option is unlikely to be widely applied because most NGHGIs are fully or partly based on direct observations (for example, national forest inventories), which cannot separate the direct human-induced effects from indirect and natural effects. The second option might be theoretically feasible but could have relevant implications in terms of incentives, fairness of mitigation efforts and compliance risk. Countries would be encouraged to invest in the monitoring of areas for which limited information exists, and in potentially protecting the carbon stocks therein: accounting would incentivise measurement and preservation. For example, in countries like Canada and Russia, fires on remote, unmanaged forest land are not suppressed as actively as on managed land, unless there is a direct threat to people or infrastructure. Furthermore, extending the area for which $CO_2$ fluxes are reported and accounted for could impact the fairness of the mitigation efforts: forest-rich countries could be incentivised to expand the area of managed forests to more easily reach emission reduction targets and carbon neutrality. However, according to the IPCC (2006), when moving unmanaged land to managed land it is good practice to describe the processes that lead to the re-categorization, i.e., countries cannot move lands in and out the NGHGI without evidence of the actual status of the land as well as of the legacy of past events (for this reason, shifting from managed land to unmanaged land is not a good practice, as the legacy effects of past management can continue for long periods). On the other hand, such a choice could imply large (and potentially uncontrollable) compliance risks for the country, associated with, e.g., permafrost thawing, large fires, etc. The concept of managed land has also been designed to reflect the intention to report and account only those fluxes that countries consider manageable. Of course, some unmanageable flux may also occur on areas considered managed by countries, which may also pose compliance risks. Related to this and following IPCC methodologies (IPCC, 2019a), countries like Canada and Australia already disaggregate emissions and subsequent $CO_2$ removals from large natural disturbances

occurring on managed land (under the assumption that fluxes compensate over time), with the aim to better isolate the anthropogenic signal on land-use emissions, and to reduce the risk that uncontrollable natural events threaten the fulfilment of the country's climate targets (IPCC, 2019a; Kurz et al., 2018). It is important to note that these natural disturbance emissions and subsequent removals are excluded from the accounting, but are reported in the NGHGI. In the future, it is likely that emissions due to natural disturbances will increase under climate change (Anderegg et al., 2020), and that the positive effects of indirect effects on the net land sink will decline (e.g., $CO_2$ fertilization will likely tend to zero under high mitigation scenarios, Canadell et al. 2021). Due to the associated compliance risk, the application of the second option (report and account for all land) could induce large and unforeseeable political risks in several countries. Yet, quantification of GHG emissions and removals on unmanaged land remains of high scientific relevance and should be encouraged.

A more realistic way forward for countries is to continue with the current approach, based on the managed land proxy specified at the country level, while investing in a number of key improvements. First, NGHGIs need to provide more transparent and traceable information on all their managed land (i.e., forest and non-forested areas), including maps of the considered areas (forest and non forest) and information on the extent to which indirect and natural effects are included. This will enable the scientific community to provide an independent (yet comparable) assessment of the NGHGI estimates, thus increasing trust in land-use flux estimates. This aspect is especially crucial in NAI countries, where the lack of specific information on managed and unmanaged areas is one of the largest knowledge gaps in the LULUCF part of most NGHGIs. In this regard, important improvements towards a more transparent and harmonized reporting are expected under the Paris Agreement's Transparency Framework (https://unfccc.int/enhanced-transparency-framework), starting at the end of 2024. Second, countries need to improve the accuracy and completeness of their NGHGIs. Many NGHGIs are still incomplete, especially for soil carbon and non-forest land categories, where observation-based estimates are often lacking. In this regard, a huge effort in capacity building for estimating land-use fluxes is needed in those developing countries with limited resources and experience in reporting, based on existing efforts and lessons learned (e.g., in the context of REDD+). This effort could involve the scientific community, e.g. in making products from Earth Observation and/or modelling directly usable by GHG inventory experts for building maps of GHG fluxes, in combination with the direct observations and statistics already available in the country. Third, the inclusion of estimates of non-anthropogenic fluxes from unmanaged lands could be included for information purposes in National Communications; while not being used for accounting purposes, it would help to better understand the responses of terrestrial ecosystems to climate change, that are crucial for assessing progress towards the goals of the Paris Agreement (Grassi et al., 2018; IPCC, 2019b).

Overall, in the short term (i.e., before 2030), it is unlikely that countries will change their approach to reporting anthropogenic land-use fluxes from managed lands, due to methodological reasons (most NGHGIs are based on direct observations, which cannot fully separate human-induced and natural effects) and policy concerns (compliance risks). In addition, any changes would first need to be included in new IPCC guidelines and approved by the UNFCCC, a process that usually takes many years. However, having more transparent,

accurate, and complete NGHGIs would already be a major achievement. As many NAI countries are in the middle of developing more sophisticated monitoring systems to comply with the requirements of the Paris Agreement, improvements can be expected in the upcoming years. Improving NGHGIs data is also critical for countries to track the impacts of land-related climate policies at national level and for updating successive NDCs. Substantial progress in reducing the gap between global models and NGHGIs can also be achieved by a different aggregation of models' results, as shown in our study. This is an easier task than changing the approach in the countries' NGHGI systems, which are based on established IPCC guidelines and UNFCCC reporting decisions.

**Conclusions**

This study confirms a substantial gap in land-use flux estimates between BMs and NGHGIs, equal to 6.7 $GtCO_2$ $yr^{-1}$ globally for the period 2000-2020, with the majority of the discrepancy occurring on forest land. For the first time, we also provide a comprehensive comparison for specific categories, such as forest land, deforestation, organic soil, and others, at the regional and country level. When BMs, which only reflect direct anthropogenic effects, are adjusted with estimates from DGVMs to incorporate the human-induced environmental change (indirect human-induced effects) on managed forests, the gap is greatly reduced at the global level (to 0.3 $GtCO_2$ $yr^{-1}$) and for most regions and countries. This confirms that the majority of the difference in land $CO_2$ fluxes between global models and country reports is not due to differences in flux estimates in a given area, but rather due to whether these fluxes are considered anthropogenic (and thus reported in NGHGIs) or natural. By making estimates of BMs conceptually and quantitatively more comparable to NGHGIs, our approach contributes to bridging the estimates of these two different communities and enables methodological improvements and consistency with global budgets that determine climate trajectories and pathways to net-zero emissions.

However, some relevant discrepancies remain, which deserve further investigation from both NGHGIs and global models. For example, the adjusted BMs' results provide a forest sink that is often greater than NGHGIs, especially in NAI countries, while in Asia, BMs estimate higher $CO_2$ emissions from deforestation and agricultural lands than NGHGIs. Our study also highlights priority areas of research for future comparisons between global models and NGHGIs, such as identifying the fluxes associated with shifting agriculture and further disaggregating the fluxes to the level of carbon pools (at least biomass and non-biomass).

Irrespective of the attribution of the net $CO_2$ flux in managed land to anthropogenic or natural drivers - which might have implications for the climate targets of countries - it is crucial for climate policy development to understand with greater confidence where this flux occurs (i.e., which country, which land use, which pools are affected), along with its temporal evolution. In this regard, future studies could test the plausibility of our estimated fluxes for managed land (i.e., a net sink of ca. -2 $GtCO_2$ $yr^{-1}$) relative to the net land-atmosphere flux

from the Global Carbon Budget (a sink of ca. -6 $GtCO_2$ $yr^{-1}$). Particularly, it remains questionable whether the difference between the two estimates occurs in the relatively small land area that has remained unmanaged. If this is the case, the individual contributions of unmanaged forest, grassland, and wetlands should be quantified. If not, it would imply that our estimated net global sink for managed land is underestimated, e.g. because the NGHGIs are incomplete and do not fully include the impact of human-induced environmental change, and/or because of the relatively simple representation of management processes by the BMs.

By harmonizing the way the anthropogenic land-use fluxes are estimated by global models and countries, this study represents an important step forward for increasing confidence in LULUCF fluxes at global, regional, and country level. While offering a blueprint for operationalizing future comparisons, our approach builds an upscaling framework that ensures greater consistency between the reporting by countries and the estimates and constraints offered by the global carbon budget. This consistency is crucial to building the necessary confidence in our monitoring and reporting systems, and therefore to support investment in land-use mitigation and assess countries' collective progress under the Global Stocktake process towards the goals of the Paris Agreement.

## Data availability

The data from this study are openly available in the online repository (Grassi et al., 2023, https://zenodo.org/record/7541525#.Y8WF8ezMJEI. 2023) including, for each country, the land $CO_2$ flux from global models (BMs and the ensemble mean of the DGVMs) and from NGHGIs for each land-use category. Furthermore, in the same repository, we made available the detailed protocol to process the DGVMs results and the map of managed/non-intact forest used in this study.

## Author contribution

G.G. led the study design, performed the analysis and wrote the first draft with the help of J.P., C.S., T.G., R.S.H., S.S., J.G.C., A.C., P.C., S.F., W.A.K., M.J.S.S. The analysis of DGVM data on non-intact forest was performed by R.A., supported by G.C.; S.F., J.M., S.R., A.R., R.A.V. and A.K. helped in the analysis of country GHG inventories. S.B., J.P., C.S., T.G. and R.S.H. provided data from BMs; S.S., P.F., S.F., E.K., D.K., J.K., M.J.M., J.N., B.P., H.T., A.P.W., W.Y., and X.J. provided data from DGVMs. All the authors provided feedback to the manuscript.

## Competing interests

The authors declare that they have no conflicts of interest.

## Disclaimer

The views expressed are purely those of the writers and may not under any circumstances be regarded as stating an official position of the European Commission or any other institution.

## Acknowledgments

G.G. acknowledges funding from the EU's Horizon 2020 VERIFY project (no. 776810). J.G.C. acknowledges the support of the Australian National Environmental Science Program - Climate Systems Hub. T.G. acknowledges support from the European Union's Horizon 2020 research and innovation programme under grant agreements #773421 (Nunataryuk project) and #101003536 (ESM2025 project). ORNL is managed by UT-Battelle, LLC, for the DOE under contract DE-AC05-1008 00OR22725. The authors thank Peter Anthoni and Almut Arneth (LPJ-GUESS model) and Sebastian Lienert (LPX model).

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

# TABLES

**Table 1.** Categories of land $CO_2$ fluxes from NGHGIs and bookkeeping models analysed in this study.

| | **NGHGIs** (see Grassi et al., 2022 for details) | **Bookkeeping models** (see Friedlingstein et al. 2022 for details) | | |
| --- | --- | --- | --- | --- |
| | | **H&N** | **BLUE** | **OSCAR** |
| **Managed forest land** | Fluxes in existing forest (including 'forest land remaining forest land', 'land converted to forest', and carbon stock changes in harvested wood products, but excluding organic soils) | Fluxes from industrial and fuel wood harvest and subsequent regrowth, product decay, plantation, recovery of forest after shifting cultivation, conversion of any type of land to forest, and fire suppression | Fluxes from wood harvest and subsequent regrowth, product decay, recovery of forest after shifting cultivation, conversion of any type of land to forest | Fluxes from wood harvest and subsequent regrowth, product decay, conversion of any type of land to forest (including recovery of forest after shifting cultivation) |
| **Deforestation** | Fluxes due to area converted from forest to other land use categories in the last 20 years, excluding fluxes from organic soils. | Fluxes due to conversion of forested land to croplands/ pastures/other lands | Fluxes due to conversion of forested land to croplands/ pastures (including emissions due to shifting cultivation) | Fluxes due to conversion of forested land to any other type of land (including due to shifting cultivation) |
| **Organic soils** | Fluxes from organic soils in various land categories (forest land, cropland, grassland) | Fluxes on peatland from external dataset (see see Appendix C2.1 in Friedlingstein et al. 2022) | | |
| **Other managed land** | Fluxes from lands not included in the above categories (e.g. cropland, grassland, wetland, settlements,and conversion non involving forests) | Fluxes due to changes in non-forest land (conversions between croplands, pastures and other lands) | Fluxes due to changes in non-forest land (conversions between croplands and pastures, clearing and regrowth of non-forest ecosystems, changes in croplands/pastu res/harvesting) | Fluxes due to harvest in non-forested land, all other conversions (between non-forested natural lands, croplands, pastures, urban lands) |
| **LULUCF net** | Sum of fluxes of the categories above, as reported in the Common Reporting Format tables | Sum of fluxes of the categories above ('ELUC net') | | |

**FIGURE CAPTIONS**

**Figure 1**. Conceptual illustration of the different approaches for estimating the anthropogenic and natural land
$CO_2$ fluxes by global models used in the Global Carbon Budget (bookkeeping models and Dynamic Global Vegetation Models, DGVMs) and by countries' National GHG inventories (NGHGIs). Bookkeeping models consider as anthropogenic only direct human-induced fluxes from land-use change, such as from deforestation, shifting cultivation, wood harvest, and regrowth after harvest or abandonment of agricultural lands. By contrast, countries in their NGHGIs generally consider as anthropogenic all the fluxes occurring on a larger area of managed forest than the one used by models, and include most of indirect human-induced effects on this area that models consider natural (i.e. the natural response to human-induced environmental changes such as increased $CO_2$ atmospheric concentration and nitrogen deposition, which enhance tree growth). NGHGIs do not consider fluxes from unmanaged lands. Note that the figure is an simplification: DGVMs can also estimate the anthropogenic flux, but here only the natural fluxes are shown (see Methods); not all NGHGIs include all indirect effects in managed land; other differences between BMs and NGHGIs exist that are not included in this figure, e.g. on the representation of forest management and forest demography.

**Figure 2**. (a) Forest map used in this study, based on maps of intact forest (Potapov et al., 2017) and non-intact forest (total forest area from Hansen et al., 2013 minus area of intact forest), except for Canada and Brazil where the NGHGI maps of managed and unmanaged forest are used (see Methods); (b-g) statistics of managed and unmanaged forest in 2015 based on NGHGIs (Grassi et al., 2022) compared to the forest map used in this study, for the world and five macro-regions (see Supplementary Table 1 and Grassi et al. 2023 for individual countries). This study uses the maps of intact and non-intact forests as a proxy for unmanaged and managed forests, respectively, except for Brazil and Canada where the country maps were available. For Russia, the tree cover threshold from Hansen et al. (2013) was adjusted to have a better match with the regional distribution of managed forest in the NGHGI.

**Figure 3**. $CO_2$ fluxes from LULUCF between 2000 and 2020 (panel a), forest land (b, including harvested wood products and excluding organic soils), deforestation (c), organic soils (d), and other fluxes (e, including cropland and grassland), from bookkeeping models (BMs) and National GHG inventories (NGHGIs). The values of BMs are those used in the Global Carbon Budget 2022 (Friedlingstein et al., 2022); values for NGHGIs are from Grassi et al. (2022), updated for DRC in this study. For organic soils, the same external dataset is used by all BMs, and their lines thus lie on top of each other.

**Figure 4**. Global land-use $CO_2$ fluxes from recent studies: BMs in the Global Carbon Budget 2020 (Friedlingstein et al., 2021) and in the IPCC 6[th] Assessment Report (IPCC, 2022); Integrated Assessment Models (IAMs) and NGHGIs in (Grassi et al., 2021); BMs in the Global Carbon Budget 2022 (Friedlingstein et al.. 2022) and NGHGIs in Grassi et al. (2022) updated for DRC in this study. On the right, the gaps between global

models and NGHGIs estimated by Grassi et al. (2021) (for the period 2005-2015) and by this study (for 2000-2020) are shown.

**Figure 5**. $CO_2$ fluxes due to environmental change (indirect human-induced and natural effects) for intact and non-intact forests from 2000 to 2020 (panel a, average of 16 DGVMs), and for non-intact forest only (b, average, and values of individual DGVMs). The DGVM simulations used here are the ones performed for the Global Carbon Budget 2022 (Friedlingstein et al. 2022).

**Figure 6**. Adjusted $CO_2$ fluxes from BMs for LULUCF (panel a) and for forest land (b), i.e. original BMs' results plus the natural sink from DGVMs in non-intact forest, compared to the NGHGIs for the period 2000-2020.

**Figure 7.** LULUCF $CO_2$ fluxes (average 2000-2020) from BMs, from the sum of BMs and DGVMs (in non-intact forest only) and from NGHGIs, for the world, Annex I countries (AI) and Non Annex I countries (NAI, panel a), for five macro-regions (b) and for 10 large individual countries (c).

**Figure 8**. Land $CO_2$ fluxes (average 2000-2020) from BMs, from the sum of BMs and DGVMs (in non-intact forest only) and from NGHGIs for the total LULUCF sector, forest land (including harvested wood products and excluding organic soils), deforestation, organic soils, and other (cropland, grassland, etc.) at global level (panel a), for Annex I countries (b), and for Non Annex I countries (c).

**Figure 9**. Land $CO_2$ fluxes (average 2000-2020) from BMs, from the sum of BMs and DGVMs (in non-intact forest only) and from NGHGIs for forest land (panels a and b, including harvested wood products and excluding organic soils), deforestation (c and d), organic soils (e and f), and other (g and h, including cropland, grassland, etc). A larger number of country-level data are included in Supplementary Table 1. Results for all countries are included in the online repository (Grassi et al., 2023).

**Figure 10**. Components of the global land $CO_2$ flux from various sources: (a) flux due to direct anthropogenic effects from BMs; (b) natural terrestrial sink, reflecting the indirect anthropogenic effects on managed forest (striped area), on unmanaged forest (green area) and on non-forest land (grey area) as decomposed in our study; (c) net land-to-atmosphere flux (sum of (a) and total area in (b)); (d) adjusted BMs' results ((a) + striped area in (b)); (e) net flux on managed land from NGHGIs (Grassi et al., 2022), updated for DRC in this study; (f) results from inversion models for managed (dashed area) and unmanaged lands (Deng et al., 2022). Estimates in columns a, b and c are from Friedlingstein et al. (2022) and refer to averages for the period 2000-2020 (like columns d and e). Estimates in column f refer to the period 2007-2017.

**FIGURES**

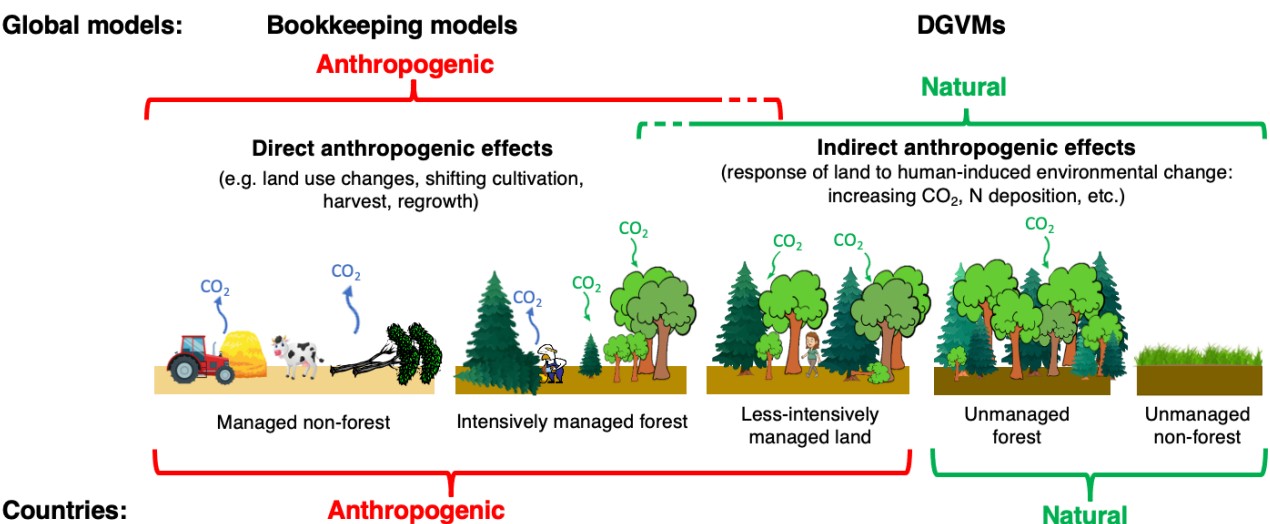

**Figure 1**. Conceptual illustration of the different approaches for estimating the anthropogenic and natural land $CO_2$ fluxes by global models used in the Global Carbon Budget (bookkeeping models and Dynamic Global Vegetation Models, DGVMs) and by countries' National GHG inventories (NGHGIs). Bookkeeping models consider as anthropogenic only direct human-induced fluxes from land-use change, such as from deforestation, shifting cultivation, wood harvest, and regrowth after harvest or abandonment of agricultural lands. By contrast, countries in their NGHGIs generally consider as anthropogenic all the fluxes occurring on a larger area of managed forest than the one used by models, and include most of indirect human-induced effects on this area that models consider natural (i.e. the natural response to human-induced environmental changes such as increased $CO_2$ atmospheric concentration and nitrogen deposition, which enhance tree growth). NGHGIs do not consider fluxes from unmanaged lands. Note that the figure is an simplification: DGVMs can also estimate the anthropogenic flux, but here only the natural fluxes are shown (see Methods); not all NGHGIs include all indirect effects in managed land; other differences between BMs and NGHGIs exist that are not included in this figure, e.g. on the representation of forest management and forest demography.

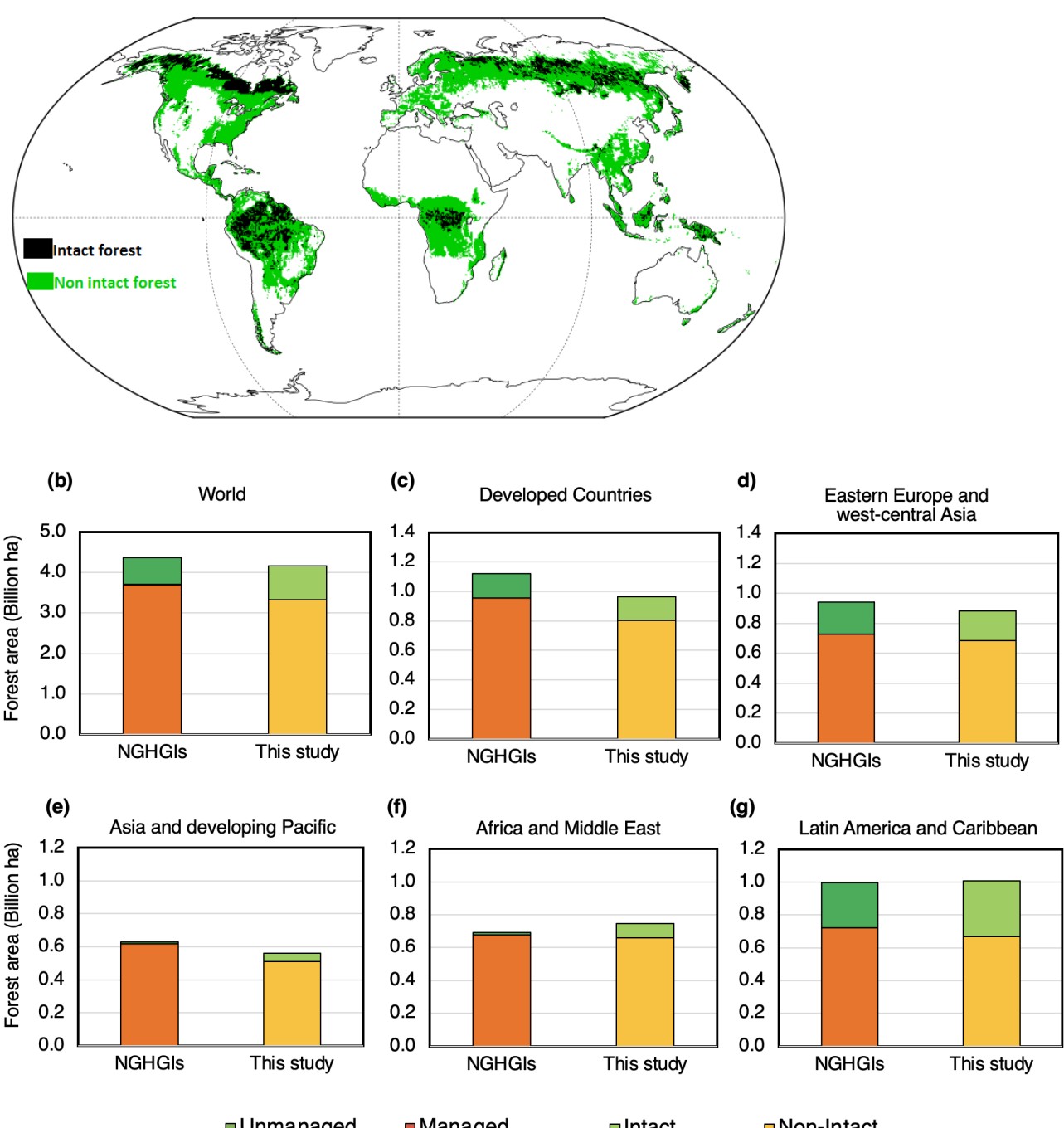

**(a)**

Intact forest
Non intact forest

**(b)** World

**(c)** Developed Countries

**d)** Eastern Europe and west-central Asia

**(e)** Asia and developing Pacific

**(f)** Africa and Middle East

**(g)** Latin America and Caribbean

Forest area (Billion ha)

NGHGIs    This study

■ Unmanaged    ■ Managed    ■ Intact    ■ Non-Intact

**Figure 2**. (a) Forest map used in this study, based on maps of intact forest (Potapov et al., 2017) and non-intact forest (total forest area from Hansen et al., 2013 minus area of intact forest), except for Canada and Brazil where the NGHGI maps of managed and unmanaged forest are used (see Methods); (b-g) statistics of managed and unmanaged forest in 2015 based on NGHGIs (Grassi et al., 2022) compared to the forest map used in this study, for the world and five macro-regions (see Supplementary Table 1 and Grassi et al. 2023 for individual countries). This study uses the maps of intact and non-intact forests as a proxy for unmanaged and managed forests, respectively, except for Brazil and Canada where the country maps were available. For Russia, the tree cover

threshold from Hansen et al. (2013) was adjusted to have a better match with the regional distribution of managed forest in the NGHGI.

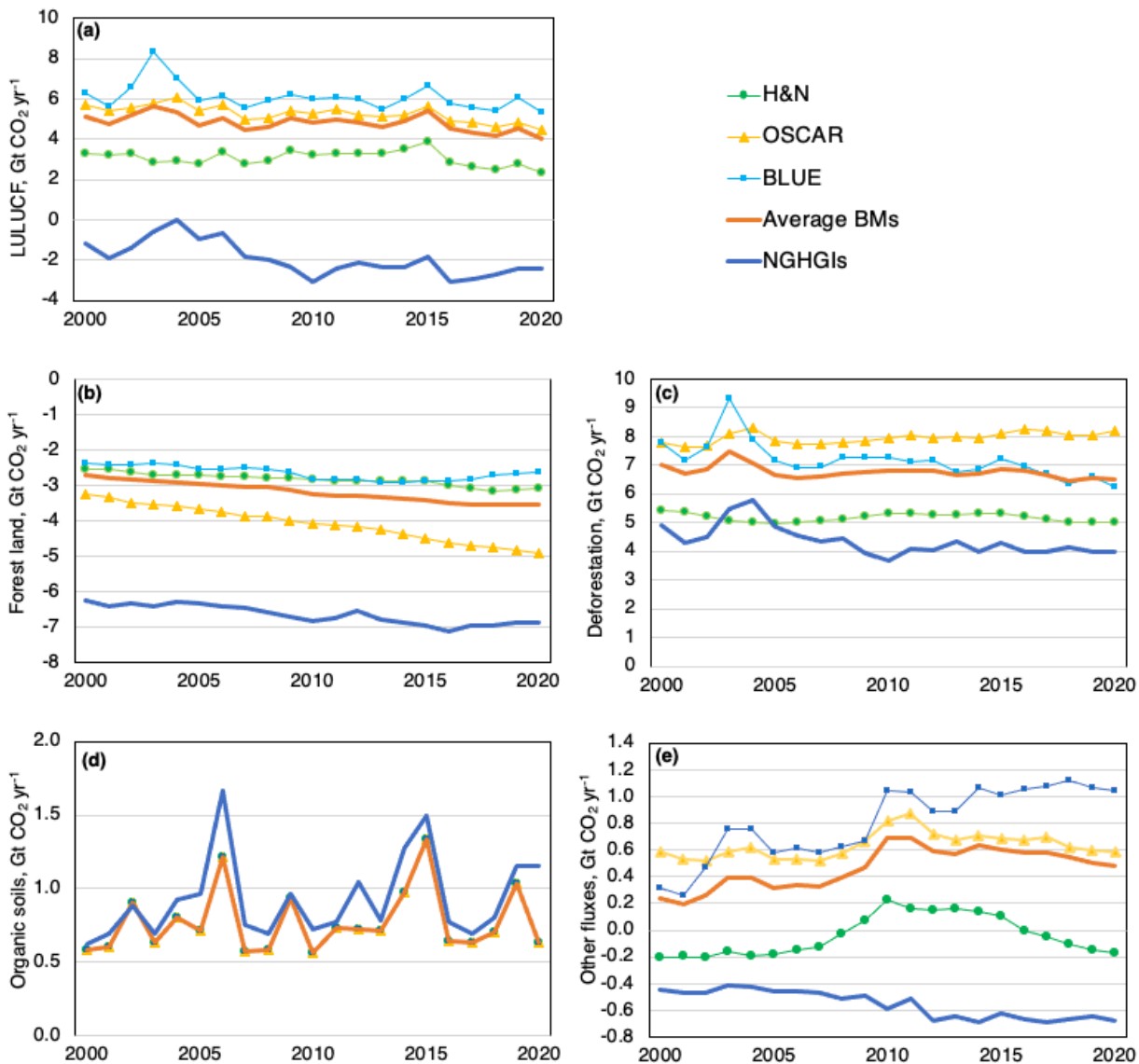

5    **Figure 3**. CO$_2$ fluxes from LULUCF between 2000 and 2020 (panel a), forest land (b, including harvested wood products and excluding organic soils), deforestation (c), organic soils (d), and other fluxes (e, including cropland and grassland), from bookkeeping models (BMs) and National GHG inventories (NGHGIs). The values of BMs are those used in the Global Carbon Budget 2022 (Friedlingstein et al., 2022); values for NGHGIs are from Grassi et al. (2022), updated for DRC in this study. For organic soils, the same external dataset is used by all

10   BMs, and their lines thus lie on top of each other.

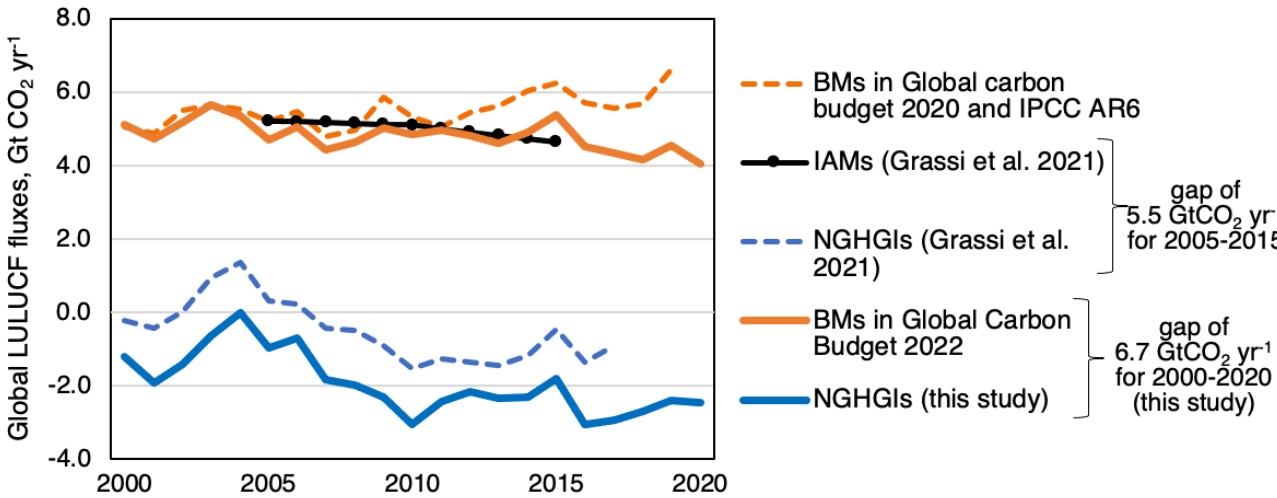

**Figure 4**. Global land-use $CO_2$ fluxes from recent studies: BMs in the Global Carbon Budget 2020 (Friedlingstein et al., 2021) and in the IPCC 6th Assessment Report (IPCC, 2022); Integrated Assessment Models (IAMs) and NGHGIs in (Grassi et al., 2021); BMs in the Global Carbon Budget 2022 (Friedlingstein et al.. 2022) and NGHGIs in this study (small update of Grassi et al., 2022). On the right, the gaps between global models and NGHGIs estimated by Grassi et al. (2021) (for the period 2005-2015) and by this study (for 2000-2020) are shown.

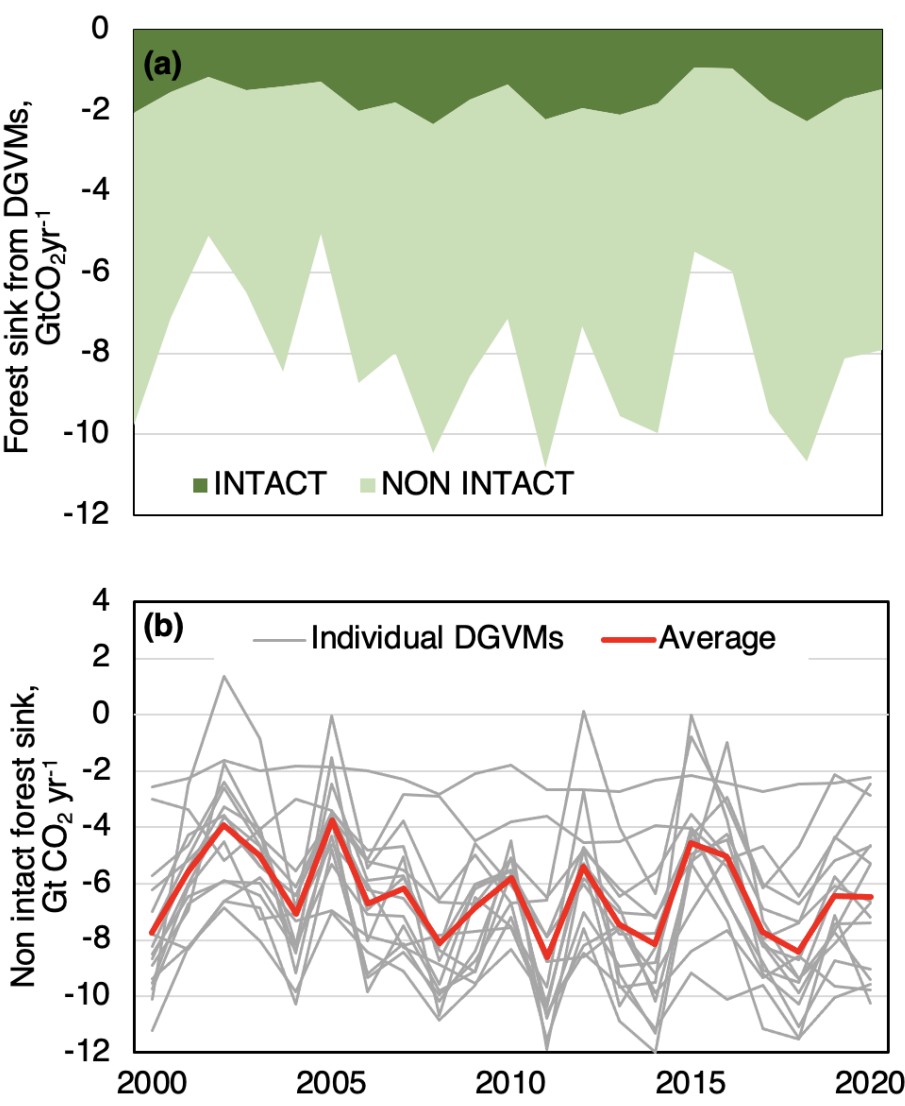

**Figure 5**. CO$_2$ fluxes due to environmental change (indirect human-induced and natural effects) for intact and non-intact forests from 2000 to 2020 (panel a, average of 16 DGVMs), and for non-intact forest only (b, average, and values of individual DGVMs). The DGVM simulations used here are the ones performed for the Global Carbon Budget 2022 (Friedlingstein et al. 2022).

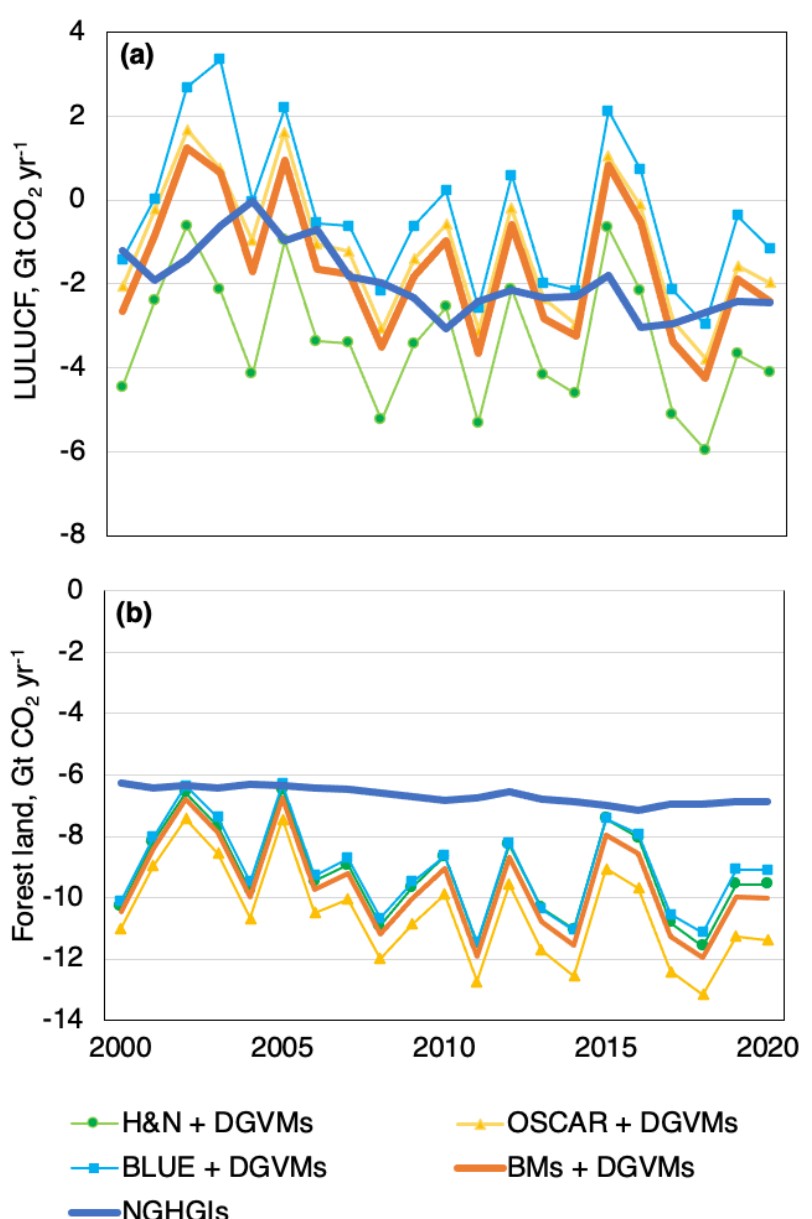

**Figure 6**. Adjusted $CO_2$ fluxes from BMs for LULUCF (panel a) and for forest land (b), i.e. original BMs' results plus the natural sink from DGVMs in non-intact forest, compared to the NGHGIs for the period 2000-2020.

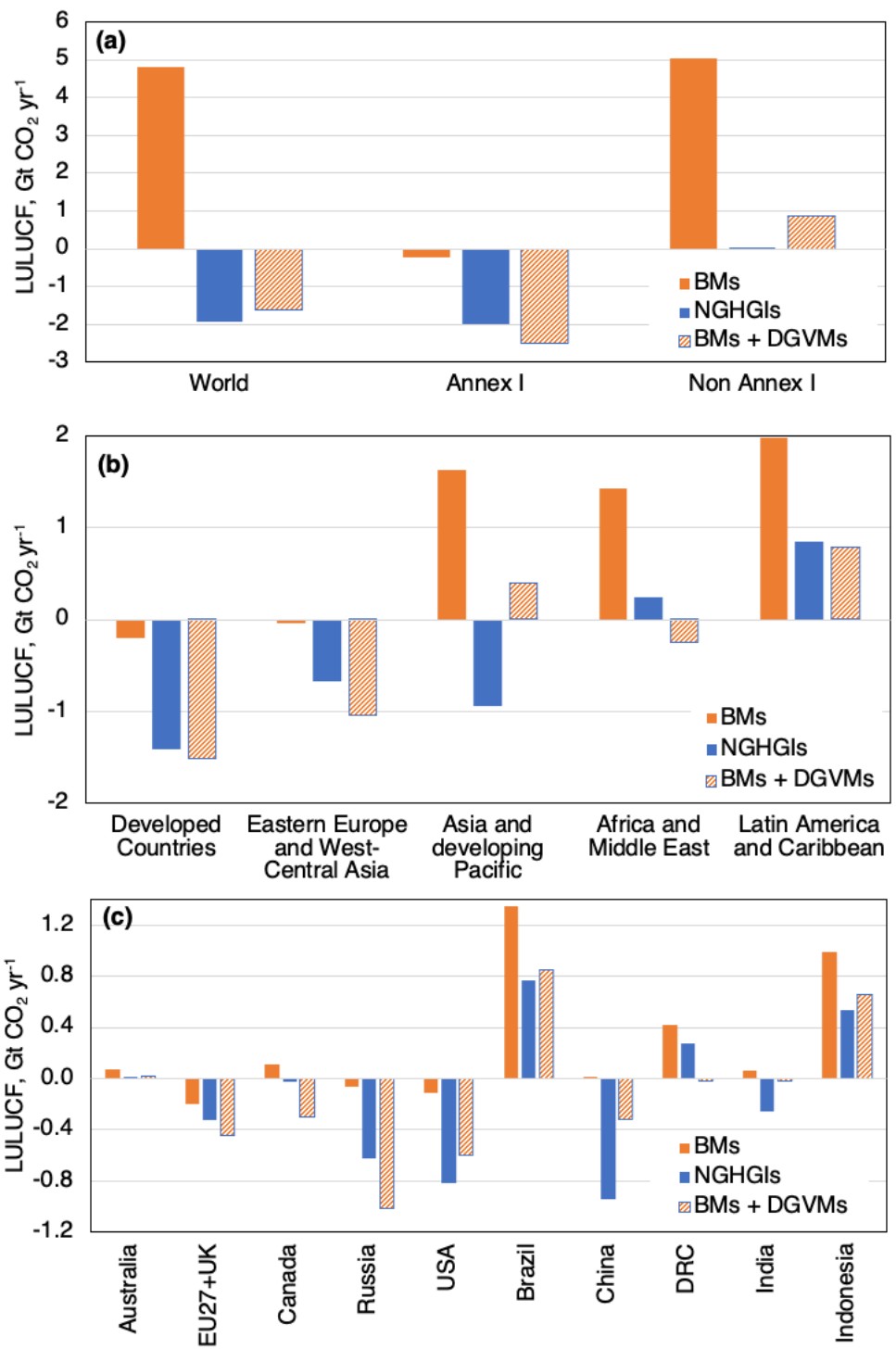

**Figure 7.** LULUCF $CO_2$ fluxes (average 2000-2020) from BMs, from the sum of BMs and DGVMs (in non-intact forest only) and from NGHGIs, for the world, Annex I countries (AI) and Non Annex I countries (NAI, panel a), for five macro-regions (b) and for 10 large individual countries (c).

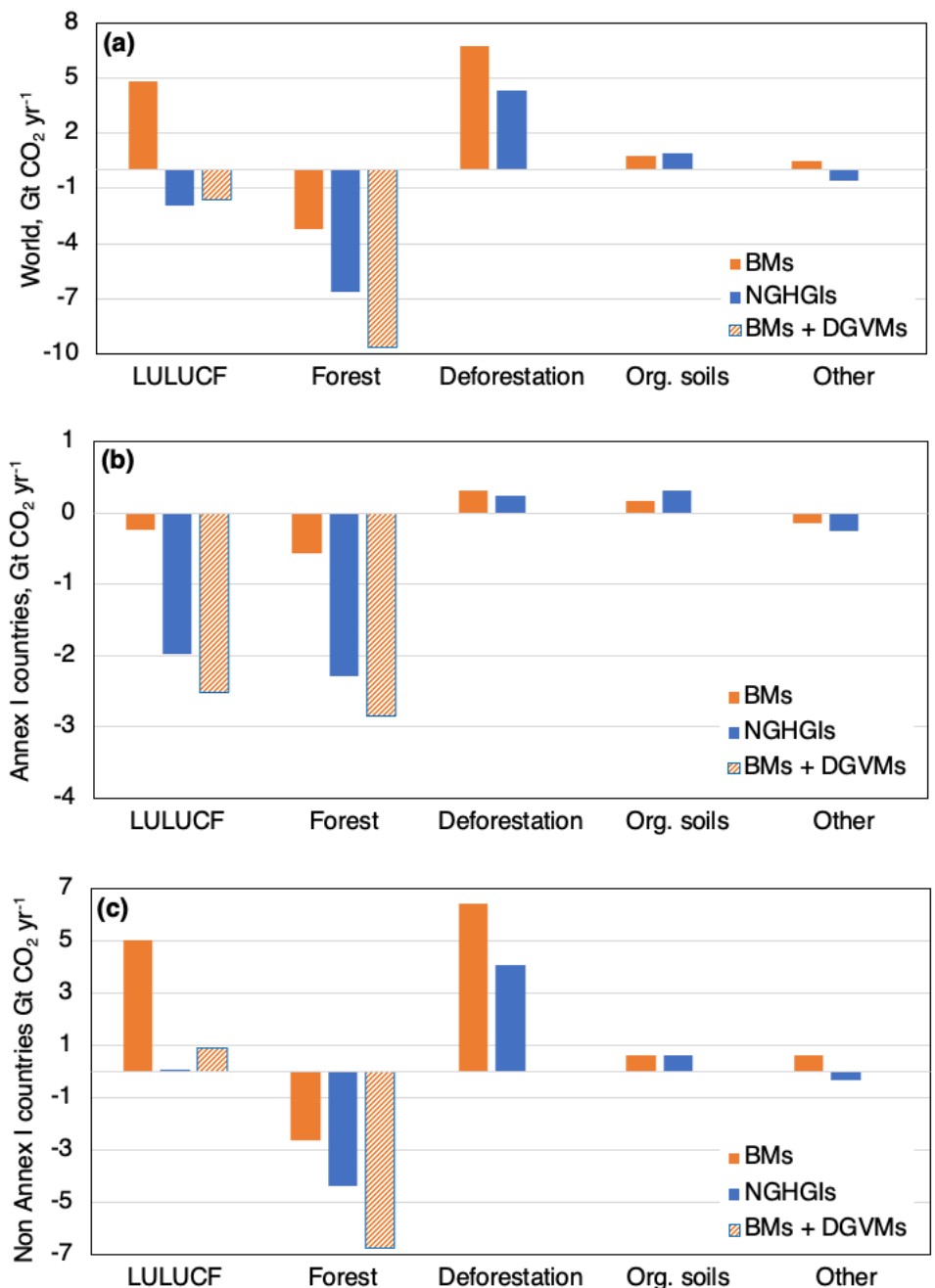

**Figure 8**. Land $CO_2$ fluxes (average 2000-2020) from BMs, from the sum of BMs and DGVMs (in non-intact forest only) and from NGHGIs for the total LULUCF sector, forest land (including harvested wood products and excluding organic soils), deforestation, organic soils, and other (cropland, grassland, etc.) at global level (panel a), for Annex I countries (b), and for Non Annex I countries (c).

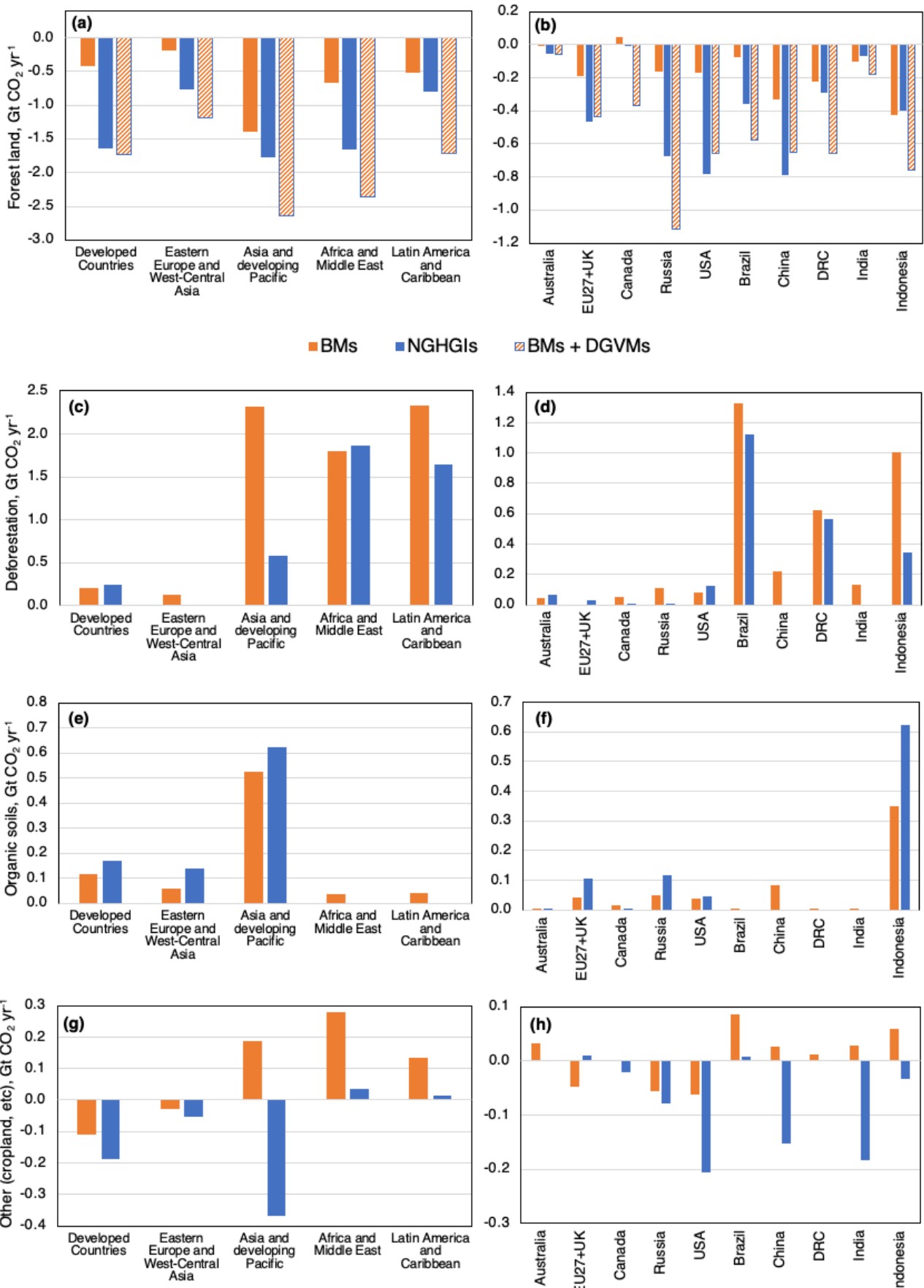

**Figure 9**. Land $CO_2$ fluxes (average 2000-2020) from BMs, from the sum of BMs and DGVMs (in non-intact forest only) and from NGHGIs for forest land (panels a and b, including harvested wood products and excluding organic soils), deforestation (c and d), organic soils (e and f), and other (g and h, including cropland, grassland, etc). A larger number of country-level data are included in Supplementary Table 1. Results for all countries are included in the online repository (Grassi et al., 2023).

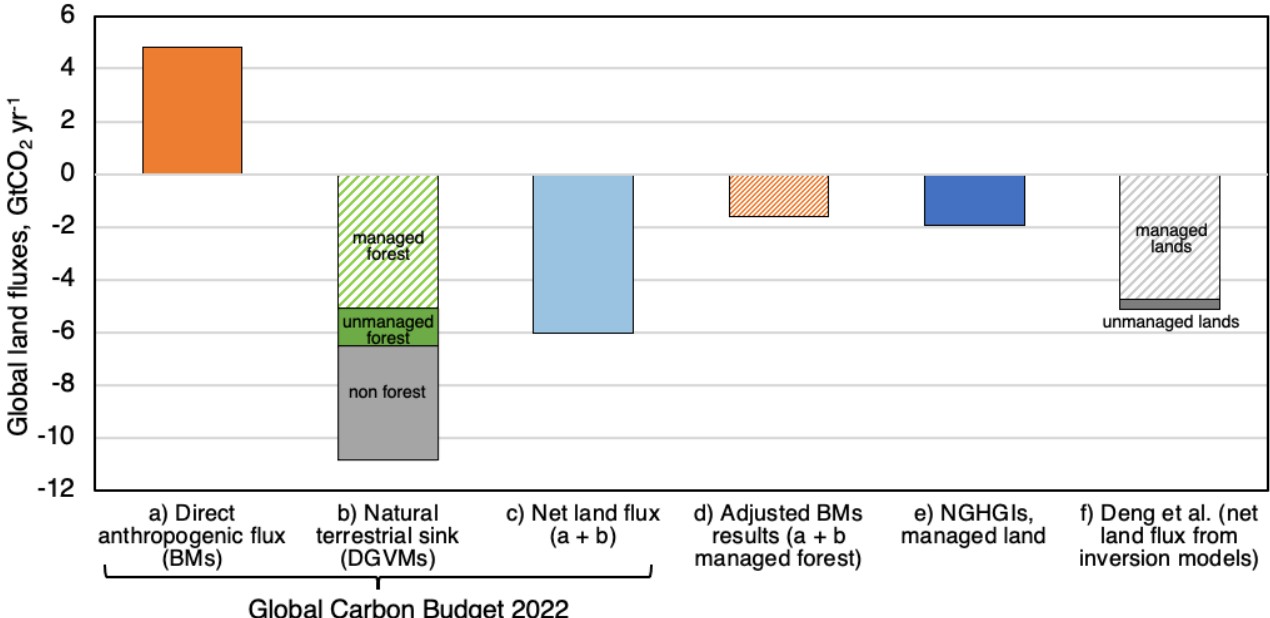

**Figure 10**. Components of the global land CO$_2$ flux from various sources: (a) flux due to direct anthropogenic effects from BMs; (b) natural terrestrial sink, reflecting the indirect anthropogenic effects on managed forest (striped area), on unmanaged forest (green area) and on non-forest land (grey area) as decomposed in our study; (c) net land-to-atmosphere flux (sum of (a) and total area in (b)); (d) adjusted BMs' results ((a) + striped area in (b)); (e) net flux on managed land from NGHGIs (Grassi et al., 2022), updated for DRC in this study; (f) results from inversion models for managed (dashed area) and unmanaged lands (Deng et al., 2022). Estimates in columns a, b and c are from Friedlingstein et al. (2022) and refer to averages for the period 2000-2020 (like columns d and e). Estimates in column f refer to the period 2007-2017.

