# Peer review of "Mapping land-use fluxes for 2001-2020 from global models to national inventories"

_Earth System Science Data, 2022_

## Author Response (AR1)

dear Editor,

We are pleased to submit the revised version of manuscript No. essd-2022-245, which includes a point-by-point response to all reviewers' comments.

We have changed the title from "Mapping land-use fluxes for 2001-2020 from global models to national inventories" to "**Harmonizing the land-use flux estimates of global models and national inventories for 2000-2020**", as we believe it is more easily understandable for a broader audience.

We have also updated our results from global models (i.e., bookkeeping models and dynamic global vegetation models) to align with the Global Carbon Budget 2022. At the time of the original submission, only GCB 2021 data were available. Although the results have slightly changed, the original message of the paper remains. This change also involved adding some additional authors.

We have made the following main changes in response to the reviewers' comments:
• We have added an online repository that includes all the data used in our study, including CO2 flux from global models and national greenhouse gas inventories for each country and land-use category. Additionally, we have made available the detailed protocol for processing global models' results and the map of managed/non-intact forest that we used, which is directly usable by other studies.
• We have further discussed the use of non-intact forest and have implemented a country-specific adjustment to the Hansen et al. (2013) forest map for Russia to improve the match with national greenhouse gas inventory data on managed forests and their regional distribution.
• We have clarified and discussed what the IPCC Guidelines for national greenhouse gas inventories include on managed land and natural disturbances.
• We have added new data and information on carbon pools, particularly soil, and on the area of managed land from both national greenhouse gas inventories and bookkeeping models.
• We have added a discussion on fluxes from shifting agriculture, which is likely an important reason for regional discrepancies between bookkeeping models and national greenhouse gas inventories, and on a possible large CO2 sink from wetlands in Russia, which is not captured by national greenhouse gas inventories.
• We have referred to and discussed the online exchange between M. Meinshasuen and S. Federici on adjusting models vs. adjusting country data.
• We have clarified the description of global models and simplified the text on 'loss of additional sink capacity', which is not central to the scope of the paper.
• We have improved and streamlined the conclusions.

We are confident that our responses address all the reviewers' main comments, and we remain available for any further clarification.

Yours sincerely,

Giacomo Grassi on behalf of all authors.

**ANSWERS TO COMMENTS and REVIEWS to Grassi et al.**

**COMMENTS by M. Meinshausen and S. Federici**

Dear Giacomo et al.

Thank you for another great paper on this topic, i.e. the mismatch between NGHGI (directly induced sources and sinks + some indirectly induced sinks) and BM and DGVMs (directly induced sources and sinks). There is one fundamental issue that I would strongly suggest to be considered. The authors suggest to adjust the BMs and DGVMs towards the NGHGI. However, the other option is of course to adjust the NGHGIs towards the BM + DGVMs by subtracting the indirectly induced sinks out of the NGHGHIs. While I assume the authors might argue in terms of the political acceptability of their approach (no country like to see somebody arguing that their official statistics have to be "corrected" somehow), there is a strong reason to perform the adjustments to the NGHGI. The remaining carbon budget under the IPCC is defined in terms of directly induced anthropogenic emissions. Thus, not adjusting the NGHGIs makes them (in aggregate) not comparable to the remaining carbon budget, nor to all the myriad of scenarios in the WG3 IPCC emission database. Thus, I would suggest the authors should rephrase some of the article by fleshing out the DELTA, and leaving it to the user, in which direction the DELTA will be applied. One could apply it to the BMs and DGVMs (as the authors suggest) or one could apply it to the NGHGIs (as I would argue is the better way to make national inventories comparable to the a broad wealth of IPCC science). Again, the ramification of the approach that the authors at the moment choose is that the non-adjusted NGHGI are not compariable to remaining carbon budgets, total CO2 emission milestones provided by the IPCC, nor net-zero CO2 timing dates. Only by "correcting" the NGHGIs, so that they reflect total anthropogenic emissions (but not induced carbon cycle changes) can one establish the comparability.

As some of the authors well know, we had a lively debate in the UNFCCC negotiations particularly in the 2000s to only report the directly-induced anthropogenic emissions and removals. As we know, the optics are better for each country, and the methodology is difficult, when one does not separate out the indirectly induced CO2 removals. Nevertheless, the concept that we keep directly induced anthropogenic sources and sinks on one side of the equation and all the indirectly-induced changes of the carbon fluxes on the other is important for the long-term integrity of the system.

I don't think that change would require any major re-calculations. It is just a question of where the DELTA is applied, i.e. I argue for it to be (optionally at least) applied to the NGHGI.

Best,

Malte

- **CC2**: 'Reply on CC1', Sandro Federici, 25 Aug 2022

  The issue is not just "political".

  Methods to deal with terrestrial C pools contributions to atmospheric CO2 concentration must address 3 fundamental requirements:

  1. **counts what the atmosphere actually sees as a consequence of human activities**. Here is worth noting that:

     o the consequences of human activities are direct and indirect, and both in the liability of human actors, and

     o it looks not sensible (not just politically) to ask to exclude the indirect effects of human activities from the mathematics that has been set to deal with the most dangerous of the indirect effects of human actions: Global Warming.

  2. **be verifiable**. This is the main challenge that models applied for reporting under the UNFCCC must address

3. symmetrically applies to the 2 sides of the same coin, CO2 removals and CO2 emissions

Then, let me note that:

- forest growth is the result of direct and indirect human-induced effects and (largely prevalent over the other two) of natural variables (and their variability, directly or indirectly affected by human actions; e.g. no rain, no forest growth).

- indirect human-induced changes in environmental conditions have impacts on GHG fluxes counted in source categories and sectors other than CO2 from forests.

Thus, shall NGHGI (not just forest land) be based on a fictitious World with standardized conditions where indirect human-induced effects are factored out?

If so:

- who is going to set such standards? (there are 25 years of scientific discussion among authors of IPCC Guidelines on this subject)

- how GHG fluxes calculated in a fictitious World would be verifiable with actual measurements?

As a very easy example, let's take a forest plantation. Who is going to establish the fraction of C accumulation to be attributed to: 1. directly to the human actions as planting/fertilizing/watering(?), 2. the indirect effects (e.g. N deposition, Global Warming) and 3. natural variables (e.g. rain), or at least their variability. And, more importantly, what a country shall count in its NGHGI in case the forest plantation does not grow as expected by counting the direct human action of planting and fertilizing and watering? shall those directly human-induced removals not realized because of indirectly-human induced causes be anyhow counted in the NGHGI? (indeed indirect effects are asked to be excluded from NGHGI so the remaining portion is the direct-human-induced which can be modelized applying standard (?) conditions) and consequently, shall CO2 emissions sourced e.g. from fossil fuel combustion be offset by such unrealized direct-human-induced removals?

Would be that what the atmosphere sees as a consequence of human actions (this is the definition of anthropogenic; without distinction between consequences reached within the purpose of the action and any other consequences caused)? Would it be considered a sensible approach?

So, considering the urgency to address global warming and the opportunity that the start of a new IPCC cycle gives, I see it urgent for models used for projections in IPCC products to be evolved for consistency with NGHGIs which, in the end, are the only that apply the guidance for good practice in estimating anthropogenic GHG fluxes as approved by IPCC. IPCC will keep moving its mission to keep enhancing its guidance for estimating anthropogenic GHG fluxes according to advances in scientific knowledge and technical capacity.

Please, do not consider this as a response by the authors (it will be given); please consider it just a personal reflection moved by the urgency to move on this subject.

**Citation**: https://doi.org/10.5194/essd-2022-245-CC2

       o   **CC3**: 'Reply on CC2', Malte Meinshausen, 26 Aug 2022

Hi Sandro,

Good to e-meet you again this way. As there are many questions in your reply, I just reply inline here (marked with >> REPLY).

Interesting discussion,

Best,

Malte
* * *
The issue is not just "political".

**>> REPLY: Agree.**

Methods to deal with terrestrial C pools contributions to atmospheric CO2 concentration must address 3 fundamental requirements:

1. **counts what the atmosphere actually sees as a consequence of human activities**. Here is worth noting that:
    1. the consequences of human activities are direct and indirect, and both in the liability of human actors, and
    2. it looks not sensible (not just politically) to ask to exclude the indirect effects of human activities from the mathematics that has been set to deal with the most dangerous of the indirect effects of human actions: Global Warming.

**>> REPLY:**

1. **I agree with the general statement of "the accounting has to reflect what the atmosphere sees".**
2. **I disagree with the interpretation of that notion, though. The atmosphere sees the full amount of direct emissions. It is then the response of the Earth System, i.e. the carbon cycle that responds to our perturbation. Netting the direct emissions with parts of the indirect responses is what "the earth system makes of it", not "what the atmosphere sees".**
3. **According to your logic (if I understand correctly), one should mix the direct anthropogenic emissions plus the natural re-distribution of carbon into its different pools, then – taking that logic to an end – we should only report emissions that sum up to the total direct anthropogenic emissions times the (time-changing) airborne fraction. In other words, if we should "take credit" for all the indirectly induced sinks, then we basically report emissions that reflect the atmospheric concentration changes. A large fossil fuel emitter could argue that 25% of its emissions are taken up by boreal forests. And 25% are taken up by the ocean. Hence, he only reports 50% of its emissions. (And yes, I know, we had countries in the UNFCCC who made that argument).**
4. **Following the same logic becomes even more odd, when we look at methane emissions, for example. If you both report emissions plus indirectly enhanced sinks (in the case of methane for example the enhanced sink via OH), then we would report zero emissions (if we choose a long enough timeframe) – given that methane has a finite lifetime. (and yes, I know, some people have even argued for that).**
5. **And is it then the boreal forest country claiming credit for the CO2, the country with the coastal ocean waters where some CO2 is taken up, or is it the fossil fuel emitter, without whose emissions we would not have the elevated CO2 emissions in the first place and hence we wouldn't have any indirectly enhanced sinks?**
6. **Or, following your logic, will boreal forest owners then start to report all the climate-change induced wildfires as their national emissions? Will Brazil claim responsibility for any Amazon dieback triggered by climate change? Those emissions are also "indirectly human-induced". But I doubt that northern boreal countries would want to claim responsibility for**

**climate-change induced wildfires, or permafrost thawing related emissions, or methane clathrate thawing in their coastal waters and forgo the "force majeure" rules.**

7. **Thus, not only does the mix of directly and indirectly human-induced emissions not add up to "what the atmosphere sees", but also leads to an asymmetric accounting, if humanity happily takes credit for some of the natural carbon sink responses to the human perturbation, and not take responsibility for the indirectly enhanced sources.**

   **Thus, the only logical reporting from a climate science point of view is to strictly try to delineate between direct anthropogenic emissions (and sinks) and indirect ones. And, following that logic, countries should report the directly human-induced emissions and additional removals, not any indirect ones.**

2. **be verifiable**. This is the main challenge that models applied for reporting under the UNFCCC must address

**>> REPLY: Agree on that one. And that is indeed a challenge in the land-use sector – for multiple reasons (as you know better than I do):**

3. **Measurement difficulty of above and below ground carbon to start with**

4. **Natural variability (i.e. drought/wet years and respiration effects etc)**

5. **And then – on the topic – separation of directly and indirectly induced carbon pool changes.**

**Thus, not bothering about the last point of course makes the measurement "more verifiable", but not necessarily "verifiable per se..", meaning that the first two points still create sufficiently big headaches that labelling land carbon pool changes as clearly verifiable remains a bit of a stretch in any case. But yes, we can operate with some verifiable activity data and emission factor assumptions etc. Luckily, your study makes the comparability and verifiability a bit easier with regards to point c.**

6. symmetrically applies to the 2 sides of the same coin, CO2 removals and CO2 emissions

**>> REPLY: So, countries should start to include climate-change induced wildfire emissions into their NGHGI? Amazon dieback? Permafrost thawing? While that would make it symmetric, it is then a question of regime stability and assigning responsibility to indirectly induced effects to the territory where these indirect effects occur.**

 Then, let me note that:

- forest growth is the result of direct and indirect human-induced effects and (largely prevalent over the other two) of natural variables (and their variability, directly or indirectly affected by human actions; e.g. no rain, no forest growth).

**>> REPLY: No disagreement here.**

- indirect human-induced changes in environmental conditions have impacts on GHG fluxes counted in source categories and sectors other than CO2 from forests.

Thus, shall NGHGI (not just forest land) be based on a fictitious World with standardized conditions where indirect human-induced effects are factored out?

**>> REPLY: Yes, for the big chunk of CO2 fertilization effects. Absolutely. And that it**

**is roughly possible is shown by your study, which is able to calculate the DELTA (i.e. the CO2 fertilization induced uptake on managed lands). That's why I think the study is a huge step forward, but one should be open about to which side the DELTA is applied. As mentioned before, if the DELTA is only applied to the BMs and DGVMs so that the NGHGI net direct + indirect emissions are taken as the "truth", then we break the connection to all the physical climate science and its milestones, like remaining carbon budgets, net-zero years etc... But we can just apply the DELTA to either NGHGI or the BM/DGVMs and easily keep the comparability. Thus, in summary, it is great to have the DELTA – and I think your study would be even better if it were agnostic to which side the DELTA is applied. The DELTA calculates the difference and that is what we need to make sense of the difference between aggregate NGHGI and BM/DGVMs.**

 If so:

- who is going to set such standards? (there are 25 years of scientific discussion among authors of IPCC Guidelines on this subject)

**>> REPLY: Yes, and we had a long history of Brazil's interventions in the 2000s on separating out indirect and direct effects – which your study is now able to do. Thus, the scientific advancement of your study is now for the first time able to estimate this DELTA more or less reliably, or not? Anyway, whether the reporting standards are set the one way or the other is, I think, a follow-up discussion. The first important thing is that we get a better handle on the DELTA (which your study does). And that DELTA can be interpreted as a "bias-correction" of the NGHGI, if one wanted to clean them of a big chunk of the indirect effects, or it can (as your study does) applied to the BM/DGVMs. As said above, I argue that your study might want to be agnostic to how the DELTA is applied.**

- how GHG fluxes calculated in a fictitious World would be verifiable with actual measurements?

**>> REPLY: As mentioned above, verifiability is a bigger issue for the landuse sector, not confined to direct and indirect effects. I would disagree with the underlying assumption that by including indirect effects we suddenly have verifiable land-use data.**

As a very easy example, let's take a forest plantation. Who is going to establish the fraction of C accumulation to be attributed to: 1. directly to the human actions as planting/fertilizing/watering(?), 2. the indirect effects (e.g. N deposition, Global Warming) and 3. natural variables (e.g. rain), or at least their variability. And, more importantly, what a country shall count in its NGHGI in case the forest plantation does not grow as expected by counting the direct human action of planting and fertilizing and watering? shall those directly human-induced removals not realized because of indirectly-human induced causes be anyhow counted in the NGHGI? (indeed indirect effects are asked to be excluded from NGHGI so the remaining portion is the direct-human-induced which can be modelized applying standard (?) conditions) and consequently, shall CO2 emissions sourced e.g. from fossil fuel combustion be offset by such unrealized direct-human-induced removals?

Would be that what the atmosphere sees as a consequence of human actions (this is the definition of anthropogenic; without distinction between consequences reached within the purpose of the action and any other consequences caused)? Would it be considered a sensible approach?

**>> REPLY: Am not quite sure I follow the last question (but that is probably really me): What do you mean by "Shall CO2 emissions sourced e.g. from fossil fuel combustion be offset by such unrealized direct-human-induced removals?". In my view that crosses over into the more fundamental "additionality" question and is**

practically also distinct between afforestation/reforestation and "managed land" areas. Anyway, I won't do your question fully justice, but just some points to consider:

- If the action of planting a tree only results in meagre carbon pool increase, because the land overall suffers from a (climate-change) induced drought, then I understand your question whether one should give anyway full "credit" for a healthy/ strong growing tree. And no, I would argue one should not, as the additional action only resulted into a meagre additional tree (no matter why the tree is meagre or not). (whether the carbon storage is truly additional over time is a separate debate).

- Anyway, that leads, I admit, to a much longer and nuanced debate on what are the truly additional carbon pool changes and whether the CO2-fertilization effect that only acts on the additionally planted forests shouldn't be counted as well into the "additional carbon pool change". And yes, I can see the argument for that (assuming that without the extra forests, the alternative ecosystem would not have shown a similar CO2 fertilization / uptake). Thus, in some respects, that indirect CO2 fertilization effect on the "additional" tree is itself "additional". It is really the "managed forests", where claiming the CO2 fertilization effect makes a big difference – those managed forests are not "additional", and hence the CO2 fertilization effect that acts on their carbon stock is not "additional". Anyway, this is – I think – an interesting pit hole for many discussions and would lead to a nuanced matrix of additional / non-additional and direct / indirect effects. And so while I agree with you that some indirect effects could indeed be taken as a credit (those that sit in the box of acting on an "additional" carbon pool, where the alternative ecosystem would not have been subject to the same CO2 fertilization effect), the majority of all the CO2 fertilization effects fall into the managed forest area, where the CO2 fertilization effect would have acted whether we "manage" the forest or not.

Anyway, that is an interesting extra discussion, but the DELTA should be reported on its own, and not per se presented as to "correct" BM/DGVM results. It could just as well be seen to correct NGHGI results.

So, considering the urgency to address global warming and the opportunity that the start of a new IPCC cycle gives, I see it urgent for models used for projections in IPCC products to be evolved for consistency with NGHGIs which, in the end, are the only that apply the guidance for good practice in estimating anthropogenic GHG fluxes as approved by IPCC. IPCC will keep moving its mission to keep enhancing its guidance for estimating anthropogenic GHG fluxes according to advances in scientific knowledge and technical capacity.

Please, do not consider this as a response by the authors (it will be given); please consider it just a personal reflection moved by the urgency to move on this subject.

**>> REPLY: I think we agree on the urgency to address global warming.**

1. **CC4**: 'Reply on CC3', Sandro Federici, 26 Aug 2022

   Hi Malte, happy to read you

   To avoid a long dialogue, let me just present three notes -that in my understanding are just pieces of evidence- and in addition let me provide a clarification on the point I see I was not clear enough.

   E1. Anthropogenic emissions and removals include both, direct and indirect emissions and removals as per IPCC Guidelines and UNFCCC requirements (since UNFCCC requires countries to apply IPCC Guidelines). *So, your considerations about direct anthropogenic vs natural redistribution are out of*

*scope; although in the last AR6 I've seen such argument surprisingly applied to $CO_2$ removals from the atmosphere that shall be discounted because ocean degassing would release back to the atmosphere a fraction of $CO_2$ removed.*

E2. The atmosphere sees all anthropogenic -direct and indirect- emissions and removals caused by human beings; thus, the national GHG inventories and the accounting stop here. *Mixing anthropogenic (direct+indirect) emissions to and removals from the atmosphere with subsequent feedback that determines a change in the GHG concentration in the atmosphere is (of course not in my logic and) out of the scope of accounting under the UNFCCC and of NGHGIs.*

E3. Conclusions within the UNFCCC (including 10 years of negotiations under KP on Brazilian proposals) and the IPCC are concordant that direct and indirect human-induced emissions and removals are both anthropogenic and cannot be separated. In my understanding, this is for the three reasons I've provided in my previous note: 1. accounting for all anthropogenic (direct+indirect) emissions and removals that the atmosphere sees as a consequence of human actions, 2. Provide symmetry in the treatment of emissions and removals, 3. Count for verifiable quantities.

UNFCCC and IPCC conclusions are based on science, and not subject to interpretations in their application (objectively applicable).

As a clarification, I was not discussing additionality (out of scope), I was just noting that discounting indirectly caused removals brings the necessity to discount indirectly caused emissions, e.g. in a forest plantation because of climate change the expected net growth does not materialize and thus instead of reporting a net C stock loss (e.g. half plantation died because of drought) a net sink is reported (indeed without indirect effects the plantation would have been growing; and it is easy to quantify the expected growth by applying yield tables). A sink that in a national accounting eventually offsets emissions from other sources.

Ciao

Ps countries report under the UNFCCC emissions from wildfires

**Citation**: https://doi.org/10.5194/essd-2022-245-CC4

- **CC5**: 'Reply on CC4', Malte Meinshausen, 26 Aug 2022

  Hi Sandro,

  Thanks.

  I think that one source of confusion is that if you mention "IPCC", you refer to the IPCC guidelines. When I refer to the "IPCC" I refer to the three Working Group reports (including the quantifications of the remaining carbon budget, net-zero timings etc.), in which anthropogenic emissions are defined, plotted and used as the DIRECT anthropogenic emissions. As I said before, this is done because this is what the atmosphere "sees" (before the Earth system reacts with CO2 fertilization and feedbacks in the carbon cycle) (and I appreciate that you have a different understanding of what the atmosphere "sees").I also do not argue that the "IPCC reporting guidelines" are wrong. My argument is simply that the DELTA between the two definitional frameworks should be quantified (as your study nicely does), but then the assumption should not be made

that accounting for purely the direct anthropogenic emissions (BM, DGVMs, and IPCC Working Group reports) is wrong and needs to be corrected. Provide the DELTA to the different communities and let them apply it either the one way or the other.

Best,

Malte

**ANSWER TO THE COMMENTS by MEINSHAUSEN and FEDERICI**

We find the discussion between Malte Meinshausen and Sandro Federici an exemplarily and extremely interesting exchange of different point of views.

While we acknowledge that the debate is still open on adjusting models vs. adjusting country data, we think that the approach taken in the UNEP EGR 2022 (see Box 2.1 https://www.unep.org/resources/emissions-gap-report-2022) may be a pragmatic interim solution. This approach suggests adjusting the models' results when the analysis is partly or predominantly focused on country or regional levels (such as in our study), and adjusting the sum of country data to the models' results when the analysis is focused on the global level. This approach ensures that country estimates are consistent with those reported by countries themselves to the UNFCCC, and that global estimates are consistent with the carbon cycle, scenarios and climate science literature used in the IPCC Assessment Reports.

Concretely, for our study, we:

- Keep the current approach of adjusting Bookkeeping models, because a large focus of our analysis is on country data.

- Add explicitly, in the online repository (https://zenodo.org/record/7541525#.Y8WF8ezMJEI), a time series 2000-2020 for each country of what M. Meinshausen calls the 'DELTA', i.e. the sink on non-intact forest derived from DGVM results, which corresponds to the adjustment applied to Bookkeeping models.

- Refer to the exchange Meinshausen - Federici explicitly in the revised discussion of our study.

**To all reviewers:**

**Anonymous Referee #1, 01 Oct 2022**

This study represents a step-forward based on previous works by Grassi et al. and the 2021 Global Carbon Budget update. It extends the reconciliation between NGHGIs and bookkeeping models on carbon fluxes over 'managed land' to cover different categories of land use (and land use change) and to cover different nations and regions. It further establishes a framework that allows such reconciliation in the future bearing in mind both methods could evolve. Then future directions for improving the method consistency and confidence in quantifying national achievements in managed land carbon sink are provided.

I appreciate having the opportunity to review this work. This can definitely clarify confusions for those who are not so familiar with this field. I suggest its publication after addressing my comments. Most of my comments are to help enhance clarity. Some comments are rather a little philosophical and the authors can consider them as suggestive to enhance their discussions if they find it useful, but not that they must be addressed in a hard way.

Thank you for the positive and very constructive comments. Please find below our replies.

Many thanks General comments:

- In the methods section: need to explain that DGVMs in S2 simulations did not explicitly simulate secondary forest, but their simulated sinks driven by environmental changes over the domain of secondary forests from another map was used, to avoid confusions. The underlying assumption is that environmental responses of carbon sink over primary and secondary forests are the same, which is not necessarily true (Lines 366-367 describes something relevant but the results there is just a coincidence I believe. DGVMs do not simulate secondary forest so any difference, if found, can only emerge from segregation of spatial grids. I suggest removing results in lines 366-367 because it makes no real sense).

  Thank you. As suggested we added in the method section that DGVMs in S2 simulations "The S2 simulations of DGVMs assume all forests to be natural and do not explicitly simulate secondary forests."

  and later "although the differences in carbon uptake by natural and secondary forests are not considered by these runs, the DGVMs gridded results capture the spatial heterogeneity of the sink in terms of different forest types, soil types, and local climate"

- The use of Potapov et al. (2017) to approximate unmanaged forest needs to be justified. The fact that it has been used in previous studies (Deng et al., 2022) does not justify its appropriateness. The most important point is, under the very loose definition of 'managed land' in the current IPCC guidelines, the domain of managed land is almost completely up to the nations and completely loses the objectivity with almost no relation to the actual state of the land whether it's managed or not. Hence the comparisons in the first two columns of Supplement Table 1 can potentially be out of pure coincidence. Note that the nations have incentives to expand the areas of managed land, whatever their real status, if they have confidence that the concerned pieces of land will be a carbon sink for a reasonably long time in the future (hence the consistency in reporting managed land is a key in NGHGIs). I don't see the advantage of using Potapov et al. compared to an alternative approach where one just uses the fraction of managed vs. unmanaged land to simply distribute the total simulated indirect carbon sink to managed land. Second, there is the conceptual inconsistency between

primary forest and the IFLs in Potapov et al. as the authors explained in their paper, and the limit of the 500 km2 minimum size, but this is rather a minor point.

On the need to justify the use of Potapov's (and Hansen's) map:

- We clarified that "a previous study (Grassi et al. 2021) found "intact" and "non-intact" forest areas being a relatively good proxy for "unmanaged" and "managed" forest areas in the NGHGIs".

- We only partly agree that "the comparisons in the first two columns of Supplement Table 1 can potentially be out of pure coincidence". While it is true that countries have a large freedom to define their managed area, we think that, in absence of better (country-specific) information, the Hansen/Potapov map represent an acceptable proxy. After all, if relevant unmanaged areas exist in a country, these are far more likely to occur on what Potapov et al. consider "intact" area (e.g.,. far from roads). In Suppl table 1 we also added country-specific information, for example on Australia, to explain/discuss any relevant mismatch between what the NGHGI counts as "managed" and what our study sees as "managed".

- With regard to "I don't see the advantage of using Potapov et al. compared to an alternative approach where one just uses the fraction of managed vs. unmanaged land to simply distribute the total simulated indirect carbon sink", we think that actually our approach (using Potapov in the absence of better information) has the advantage that the DGVMs gridded results capture the spatial heterogeneity of the sink in terms of different forest, climate and soil types. This is added in the text (see above).

- On the "conceptual inconsistency between primary forest and the IFLs in Potapov et al. and the resolution used by DGVMs (0.5 degree), we agree, but given the data from DGVMs, there is not much we can do. However, in the revised manuscript, we made available the combined global map that we used, at two resolutions (0.5 and 0.05 degrees), to allow other studies to apply even a greater resolution than the one we used with DGVM data.

- To address, at least in part, the reviewer's concern, we developed a specific map of managed/unmanaged forest for Russia, which is one of the countries with the largest areas of unmanaged forest. Although a digital map of the managed forests considered by the Russian NGHGI is not available, the Russian GHG inventory team shared with us specific maps for each administrative region. These maps indicate the fraction of forest and the relative share of managed/unmanaged forests in each administrative region. These maps were used to guide an adjustment of the threshold used in the Hansen's forest map, aimed at obtaining a better match at regional level between our area of managed forest and the one used in the NGHGI. Specifically, the use of a threshold of 10% in tree cover (rather than 20% as for the other countries), combined with the same approach applied for the other countries, allowed us to obtain values of managed forest area which resemble reasonably well the Russian regional maps (see new Supplementary figure 2) and match well the total managed forest area reported for the whole country (666 Mha in our map, 686 Mha in the NGHGI). Although our approach does not produce an exact map of the managed forest used in the NGHGI, we think it represents a step forward in representing the spatial allocation of Russian managed forests. The resulting map for Russia was then combined with the country-specific maps for Canada and Brazil and the Hansen/Potapov map for the remaining countries, to produce a global map of managed forests. Since Russia, Brazil and Canada represent the three countries with the largest area of unmanaged forest globally, our approach minimizes the impact of using the Hansen/Potapov map. The combined global map is made available online (https://zenodo.org/record/7541525#.Y8WF8ezMJEI) at two resolutions (0.5 and 0.05 degrees), to allow other studies to apply the same approach. Given the increasing interest by

the scientific community in verifying data from NGHGIs, we think this new global map represents a useful added value compared to the original manuscript.

- The separation between forest carbon sink and deforestation fluxes is nice but also comes with uncertainty. The key is fluxes relate to gross deforestation might have not been reported by nations and the area undergoing gross deforestation depend on the spatial resolution of the land cover data used. The discussions and associated uncertainties in this respect need to be strengthened.

Thanks, we agree. As suggested, we strengthened the discussion with regard to shifting agriculture and deforestation, in two points:

"While the separation of BMs' results into various land categories helps the comparison with the NGHGIs, an important source of uncertainty (especially for NAI countries) is how the fluxes from shifting agriculture are allocated, i.e., if they are placed into forest, deforestation, or other. Specifically, in this study BLUE and OSCAR allocate emissions from shifting agriculture under "deforestation" and any subsequent removals under "forest" (e.g. for OSCAR, this corresponds to +3.5 $GtCO_2$ $yr^{-1}$ under deforestation and -2.5 $GtCO_2$ $yr^{-1}$ under forest for the period 2000-2020); H&N allocates emissions from shifting agriculture under "deforestation" only after the first conversion occurs (this corresponds to about +1.1 $GtCO_2$ $yr^{-1}$ for the period 2000-2020), and thereafter the emissions and removals (overall a small net flux) are allocated to "other fluxes". The quantitative importance of shifting agriculture for $CO_2$ fluxes is also confirmed by Harris et al. (2021). For NGHGIs, it is often unclear if and under which categories the fluxes due to shifting agriculture are reported. While the above difference may help to explain the larger emissions from deforestation in BMs than in NGHGIs, and also the larger forest sink in the adjusted BMs (i.e. including the natural sink estimated by DGVMs for the managed forest area) than in NGHGIs, the lack of reliable information from most NAI represents one of the biggest sources of uncertainty in our comparison at the level of individual land-use categories"

"Estimates of deforestation fluxes from models strongly depend on the underlying datasets, including the spatial resolution of the land cover data or statistics used (Winkler et al. 2021). For example, a recent study (Ganzenmüller et al. 2022) concluded that deforestation emissions based on high-resolution activity data substantially lower the previously estimated emissions using the LUH2 datasets (used here by BLUE and OSCAR). On the other hand, some NGHGIs do not report emissions from gross deforestation (for example, China and India)."

- I also noted the lively discussions between Malte Meinshausen and Sandro Federici. While there is perhaps no need to provide the corrections due to indirect effects because readers who understand this paper can easily obtain them by using the supplementary Table 1. But I suggest authors enhance the discussions regarding the potential leakages in current IPCC guidelines. That is, when the pervasive indirect effects of increasing CO2 show as a sink-enhancing term over actually unmanaged land, the nations have incentives to claim these lands as managed lands and the associated carbon sinks as the national contributions to mitigate climate change. But when the same indirect effects show as a carbon source term in forms of growing wildfires or large-scale forest dieback, the nations will have incentives to say these are natural disturbances and are not national liabilities.

With regard to the suggestion by Meinshasuen, we now added all our data online (https://zenodo.org/record/7541525#.Y8WF8ezMJEI). This online repository includes, for each country, the $CO_2$ flux from global models (BMs for each land-use category, and the ensemble mean of the DGVMs with the sink in intact/non-intact forest, i.e., the adjustment

applied to BMs) and from NGHGIs (for each land-use category). Furthermore, in the same repository, we made available the detailed protocol to process the DGVMs results and the map of managed/non-intact forest that we used.

With regard to the potential leakages in current IPCC guidelines -Volume 4, Chapter 3 of the 2019 Refinement to the 2006 IPCC Guidelines on National Greenhouse Gas Inventories - we clarified that:  "Furthermore, extending the area for which $CO_2$ fluxes are reported and accounted for could impact the fairness of the mitigation efforts: forest-rich countries could be incentivised to expand the area of managed forests to more easily reach emission reduction targets and carbon neutrality. However, according to the IPCC (2006), when moving unmanaged land to managed land it is good practice to describe the processes that lead to the re-categorization, i.e. countries cannot move lands in and out the NGHGI without evidence of the actual status of the land as well as of the legacy of past events (for this reason, shifting from managed land to unmanaged land is not a good practice, as the legacy effects of past management can continue for long periods). On the other hand, such a choice could imply large (and potentially uncontrollable) compliance risks for the country, associated with, e.g., permafrost thawing, large fires, etc. The concept of managed land has also been designed to reflect the intention to report and account only those fluxes that countries consider manageable. Of course, some unmanageable flux may also occur on areas considered managed by countries, which may also pose compliance risks. Related to this and following IPCC methodologies (IPCC 2019a), countries like Canada and Australia already disaggregate emissions and subsequent $CO_2$ removals from large natural disturbances occurring on managed land (under the assumption that fluxes compensate over time), with the aim to better isolate the anthropogenic signal on land-use emissions, and to reduce the risk that uncontrollable natural events threaten the fulfilment of the country's climate targets (IPCC 2019a, Kurz et al. 2018). It is important to note that these natural disturbance emissions and subsequent removals are excluded from the accounting, but are reported in the NGHGI. In the future, it is likely that emissions due to natural disturbances will increase under climate change (Anderegg et al. 2020), and that the positive effects of indirect effects on the net land sink will decline (e.g., $CO_2$ fertilization will likely tend to zero under high mitigation scenarios, Canadell et al. 2021). Due to the associated compliance risk, the application of the second option (report and account for all land) could induce large and unforeseeable political risks in several countries. Yet, quantification of GHG emissions and removals on unmanaged land remains of high scientific relevance and should be encouraged"

Below are some minor comments:

Lines 86-88: we know that these differences reside in their respective concepts and are unlikely resolved by 'new observations and platforms.' Please rephrase.

Thanks, we slightly rephrased it.

Line 933-934: "direct human-induced effects" should be "direct land use change effects", land-use change includes shifting cultivation, harvest and regrowth due to abandonment or afforestation/reforestation. Need to rephrase here.

We rephrased as "Bookkeeping models consider as anthropogenic only direct human-induced fluxes from land-use change, such as from deforestation, shifting cultivation, wood harvest, and regrowth after harvest or abandonment of agricultural lands."

Figure 1: strictly speaking there are only unmanaged lands if trendy S2 simulations are meant here. Need to also state in the methods that environmental effects on managed land are treated equal as those on unmanaged lands, which is a critical assumption in the paper but not necessarily true.

As suggested above, we added that:

" The S2 simulations of DGVMs assume all forests to be natural and do not explicitly simulate secondary forests."

and later "although the differences in carbon uptake by natural and secondary forests are not considered by these runs, the DGVMs gridded results capture the spatial heterogeneity of the sink in terms of different forest types, soil types, and local climate"

Line 112: "global models": do you mean bookkeeping models? As trendy S3 is not used throughout the whole study, it's better to be specific.

Thanks, we modified the text and now specify that we refer to Bookkeeping models

Line 125: this gives a sense that DGVMs simulate secondary forest. But in Grassi et al. (2018), only the spatial map of secondary forests (from LUH2?) were used and applied on DGVM simulated carbon sinks of intact land, no? Please correct to provide a more accurate description.

Thanks, we changed into "the CO2 fluxes from nine DGVMs were filtered with a map of secondary forests and added to the fluxes from one BM"

Line 128: "proxy for managed" => should be proxy for environmental effects on management forest.

Thanks, we followed the suggestion

Line 178: "natural land cover changes (Sitch et al., 2015)", I guess it should be "natural vegetation dynamics". The authors should mean land cover changes that are driven by environmental changes rather than human-induced land use.

Indeed, we used "natural land cover changes" in contrast to land-use-induced land cover changes. We have replaced this by "natural vegetation dynamics in response to environmental changes" to be even clearer.

Line 179: 'anthropogenic fluxes' and line 192 'anthropogenic CO2 emissions'. These should be both 'land use change emissions', not to be confused with 'anthropogenic CO2 emissions' in lines 93 and 95.

We have deleted this sentence in response to reviewer 3's comments, which made us slightly re-write this section.

Line 191: what do you mean by 'forest thinning from below'?

We deleted that, it is not strictly needed.

Line 283-286: Note that simulated natural carbon sinks over non-forest land account for ~ 1/3 by DGVMs (Table A8, 2022, ESSD; also shown in Fig. 10 in this paper). According to the approach used here, I guess some of this non-forest natural sink is ascribed to secondary forests. This might need clarification. This points needs to be considered also relevant with lines 310-315.

We now explain that the flux over non-forest land is high because it includes the loss of additional sink capacity at the end of the methods section.

Lines 374-377: the discussions here comparing the adjustments from a single model and from an ensemble of models seem not having a lot of insights for me. The agreement might be just out of coincidence. I suggest removing it.

Ok, we deleted the sentence, as suggested

Lines 384-391: The authors can just explain that the simulated natural sink per forest area was used from DGVMs so that the effects of hypothetical forest areas at preindustrial conditions were filter out, which would be easier to understand.

We have consolidated our discussion of the loss of additional sink capacity and moved this part to the methods section. This shortened our explanation overall.

Line 460-461: I guess there is also the difference in gross vs. net land use change between those accounted for in BMs and by GHGIs, not limited to shifting cultivation.

Agree. We added a sentence mentioning gross deforestation possibly underestimated by NGHGIs, see above.

Line 566: "NGHGIs typically report estimates of gross deforestation" => Are there evidences supporting this? At least this seems not the case for China and India because they do not show any deforestation flux. Reporting gross vs. net changes in forest area in the category of deforestation could be an important source of uncertainty, because gross forest change typically depends on the spatial resolution of the underlying land cover data that are used to derive land cover change. (https://www.nature.com/articles/s41467-021-22702-2)

Thanks. We added a sentence mentioning gross deforestation possibly underestimated by NGHGIs (e.g. China and India), see above. We also added ref to Winkler et al.

Line 559-560: Could it be an option for the land use change segregations used in BMs to move close to IPCC guidelines? A greater disaggregation in BMs will for sure better, but does not necessarily lead to better comparison with GHGIs and allow avoiding the cross-walking like in Table 1. The large discrepancy in the category of 'other lands' (Fig. 9) probably points to potential underlying mismatch in land use change category between these two approaches.

Thanks. We think that the four categories applied here are already a good step forward, and indeed they have been applied also in the Global Carbon Budget 2022 released after we have first submitted our paper. We have thus substantially shortened this paragraph, acknowledging the progress meanwhile made to move BMs' disaggregation closer to IPCC guidelines' categories.

On "other", we agree that the problem may partly be in the system boundaries: for example, NGHGIs are expected to report "cropland remaining cropland" and "grassland remaining grassland" (this is done by most Annex I countries, but only by very few Non Annex I countries), while BMs only estimate conversions (e.g. cropland to grassland and viceversa). This is partly already discussed, e.g. when we say that "While this may be partly due to the fact that BMs estimate only land-use changes for agricultural lands  (e.g., grassland conversion to cropland and not 'cropland remaining cropland'), the large sinks reported by India (for cropland) and China (for cropland, grassland, and wetlands) are not well documented."

Line 586-587: with climate warming as a directional change, the assumption that climate-driven changes or variations in disturbance will cancel to a net-zero effect over time will unlikely hold. This comes back to the discussions between Malte Meinshausen and Sandro Federici (cf. the open discussion online: https://essd.copernicus.org/preprints/essd-2022-245/). Both being as directional changes, carbon sink enhanced by CO2 concentration increase are accounted as national contributions to carbon sink, but carbon sources enhanced by climate change (e.g. wildfires) were left out.

As explained above IPCC Guidelines do not allow to exclude from the 'reporting' any emissions estimated applying the managed land proxy from the NGHGI - the exclusion could be made by countries during the accounting.

The assumption that the impact of disturbances average out over time is based on the "equilibrium" principle of ecology. No better alternative was found within the IPCC Guidelines. This however may change in the next future and would require additional methodological work by the IPCC.

Nevertheless, to clarify that emissions are not discretionally excluded from NGHGIs, the following revision is implemented: "It is important to note that these natural disturbance emissions and subsequent removals are excluded from the accounting, but are reported in the NGHGI"

 Line 590-592: completely agree with this.

Thanks

Finally, it should be noted that we updated the results to make it consistent with the Global Carbon Budget 2022 (at the time of the original submission, only GCB 2021 data were available). While results changed a bit (i.e., now we have a better match between BMs and NGHGIs at the level of LULUCF, but a less good match for forest land and deforestation), the message of the paper does not change.

**Anonymous Referee #2, 04 Oct 2022**

The paper provides an interesting advancement of the previous analysis of the same authors on the mismatch between national inventories and global models used for estimating the emissions of the land sector. The paper is well structured and accurate in the analysis and clarifies many of the possible reasons of mismatch between statistics and estimations, providing also perspectives for the reconciliation of these differences. Although the comparison is very detailed and accurate, there is no information on the different pools that are considered in the different sources considered (NGHGI, BM and DGVM) and whether this can be a source of mismatch between estimations. This should be clarified in the method section as it is a potential source of discrepancy (potentially also adding this information in Table 1).

Thank you for the positive and very constructive comments.

On carbon pools, we agree that more information was needed.

In the Methods, we added new information for models (both BMs and DGVMs) and NGHGIs.

For models we simply added that "Estimates from both BMs and DGVMs include all carbon pools."

For NGHGIs, we added: "In terms of reported carbon pools, the situation varies depending on the country and the land category. The IPCC guidelines (2006, 2019) distinguish living biomass (above- and below-ground), dead organic matter (dead wood and litter), soils (mineral and organic) and harvested wood products (sometimes referred to as a separate category rather than a carbon pool). The vast majority of AI countries report the $CO_2$ fluxes from the carbon pools in case of land-use changes (e.g. forest converted to settlements, cropland converted to forest, grassland converted to cropland), and from the most important carbon pools in case of land uses that remain unchanged (e.g., biomass in forest land remaining forest land, soil in cropland remaining cropland). The NAI countries typically report the $CO_2$ fluxes from living biomass on deforestation and, in the vast majority of cases, on forest land. For the other pools the situation is less clear. Dead organic matter, mineral soils and harvested wood products are reported by the largest NAI countries (including Brazil, China, India, Indonesia, Mexico) but are often not considered by other NAI countries. $CO_2$ fluxes from organic soils are reported only by a few NAI countries (e.g., Indonesia)"

Furthemore, we extracted the available data and made a preliminary comparison, in the Result and Discussion, we added:

"With regard to the allocation of fluxes to the various carbon pools, the comparison between global models and NGHGIs is hampered by different definitions of carbon pools and by incomplete estimations by NGHGIs (especially for NAI countries). Nevertheless, based on the available data, mineral soils do not seem to represent a major source of difference in land use $CO_2$ fluxes between global models and NGHGIs, at least in forest land. According to NGHGI data from AI countries for the category forest land, the vast majority of the forest sink is reported in the living biomass, with mineral soil and dead organic matter representing, respectively, 7% and 11% of the total net sink for the period 2000-2020 (excluding organic soils). This information is broadly in line with the results from global models. Data from the TRENDY v11 dataset (for nine DGVMs only), for example, show that the sink in forest soils represents about 10% of the overall forest sink in AI countries during the same period. For all land uses, the BMs' results indicate a net source from mineral soils (about 1.5 $GtCO_2$ $yr^{-1}$ from BLUE and 0.6 $GtCO_2$ $yr^{-1}$ from H&N for 2000-2020), with emissions associated with land-use changes (mostly deforestation) and a small sink in forest land. Overall, we argue that a more comprehensive analysis of fluxes in different carbon pools should be prioritized in future studies comparing global models and NGHGIs, along with analyses of possible lateral fluxes that might be overlooked by both BM and NGHGIs."

In the Conclusions:

"Furthermore, our study highlights priority areas for future comparisons between globals models and NGHGIs, such as indentifying in the NGHGIs the fluxes associated with shifting agriculture and - in both global models and NGHGIs - disaggerating further the fluxes to the level of carbon pools (at least biomass and non- biomass)."

Overall the assessment is very positive, I therefore recommend the publication with minor revisions.

Specific comments:

-Lines 94- 95 Missing quotation to the sentence "BMs estimate that land use is a net source of CO2 globally, mainly due to deforestation, equal to around 12% of total global 95 anthropogenic CO2 emissions."

Thanks, now added (Friedlingstein et al. 2022)

Line 108 Suggest to add the reference to Figure 1 at the end of the para

Since that para also include natural effects (sensu IPCC 2006), which are not represented in the figure, we prefer keeping the reference to figure 1 few lines below.

Line 112 "The main conceptual difference is that global models consider those forests as managed that were subject to recent harvest and have not yet regrown to pre-harvest stock level…." With global models are you referring to both DGVM and BM? Otherwise please specify.

Thanks, we clarified that here we refer to Bookkeeping models.

Lines 322-25: The sentence is not clear, I suggest to reformulate it.

In response to this and the other reviewers' comments we have consolidated the explanation of the loss of additional sink capacity at the end of the method section and clarified this statement there.

Line 374-377: This paragraph refers to an "adjustment" but I think it is more correct to refer to a "difference" or "gap" as the number represent a difference between estimations, while the adjustment is done when you want to make one of the two comparable to the other (e.g. line 378)

Thanks. We modified as "The sink in non-intact forest estimated by Grassi et al. (2021), using one DGVM only, was equal to -5.0 $GtCO_2$ $yr^{-1}$ for the period 2005-2020 (Supplementary Table 8, Grassi et al. 2021). In this study, for the period 2000-2020, 16 DGVMs estimate a sink in non-intact forest of -6.4 $GtCO_2$ $yr^{-1}$. In both cases, the adjustments based on these sink estimates reconcile most of the gaps identified between the anthropogenic land-use $CO_2$ flux estimated by NGHGIs and by global models (either IAMs or BMs)"

468 – missing space between "thatBMs"

Thanks, corrected

595 – not clear why a the inclusions to natural terrestrial sink would lead to a "double-counting" perhaps to a wrong attribution of the natural fluxes to anthropogenic causes?

We deleted that sentence.

Table 1: add a reference to the pools included in the different sources. On Organic soils: Oscar uses the same dataset as BLUE, I guess it is not the case of H&N. I suggest to include the source of the datasets for the three BMs.

Thanks, done.

In the row LULUCF net, last three cells (related to BMs): instead of repeating the same information, the information could be included once, merging the three cells (also in other cases above where the same approach is used by different models)

Thanks, done.

Finally, it should be noted that we updated the results to make it consistent with the Global Carbon Budget 2022 (at the time of the original submission, only GCB 2021 data were available). While results changed a bit (i.e., now we have a better match between BMs and NGHGIs at the level of LULUCF, but a less good match for forest land and deforestation), the message of the paper does not change.

**Anonymous Referee #3, 20 Oct 2022**

Review comments: Mapping land-use fluxes for 2001-2020 from global models to national inventories

I have been following this work for some years, and I am glad to see an article which describes the methods in more detail. Though, more detail leads to more questions! Overall, I think this is a well written article that makes the necessary data descriptions (with some potential modifications), and I do not see any major barriers to publication. The detailed method description does make me reflect on various historical assumptions (not made by the authors, but further locked in by the authors). In a sense, the authors mechanically go through a method to bridge the gap between different definitions, ok for ESSD I guess, but I think there could be a little more reflection on the appropriateness of past decisions and therefore potential pathways forward to avoid the 'gap' (the authors do some of this already in the discussion).

Thank you very much for the positive and constructive comments, and for the thoughtful and stimulating reflections and questions. In general, we agree with most of these reflections. Of course, as the reviewer also noted, there are no easy answers to all the questions. Below and in the revised manuscript, we try to reflect upon these inputs.

In a sense, the paper provides an ad-hoc "fix" to a problem that sort of should not exist. Of course, there may be good reasons to have different definitions, and therefore a method to bridge those definitions. Though, at some stage, one may start to question the definitions and whether they are appropriate for the times. Science evolves, as does policy. In both cases, BM and UNFCCC, are wedded to decisions made in the past. The authors lock this in with comments like "unlikely countries will change". We tend to hold on to those decisions as hard as we can, even if the justification is rather weak!

Key questions that come to mind: Is it appropriate to use BMs in the carbon budget? How can the Loss of Additional Sink Capacity not be included in the carbon budget?

Why can countries continue to have such inconsistent definitions of managed land? Is managed land still a relevant proxy for anthropogenic? If BMs included indirect effects (such as in DGVMs) and UNFCCC had data based definitions of 'managed' land, the gap may diminishe significantly? Ok, I don't expect the authors to solve these issues, but their method and dataset puts them in a unique position to comment on these issues. The UNFCCC is based on IPCC guidelines, and IPCC guidelines are also informed by IPCC ARs, so if we have better science, or can do things differently and better, that should be communicated, not just assume BMs or UNFCCC will never evolve.

Thanks, we broadly agree (but of course our paper cannot address all these points)

On models: given the combination of BMs and DGVMs can closely approximate the NGHGI definition, implementing the indirect effects in BMs would provide little added value (but of course it can be explored in future studies).

On countries, we added: "Overall, in the short term (i.e., before 2030), it is difficult that countries will change their approach to reporting anthropogenic land-use fluxes from managed lands only, due to methodological reasons (most NGHGIs are based on direct observations, which cannot fully separate human-induced and natural effects) and policy concerns (compliance risks)". We also reinforced the text before this, with the aim to provide more robust arguments to what countries may or may not likely do in the future.

"We acknowledge the open debate on how to reconcile global models and NGHGIs – see the instructive discussion between M. Meinshasuen and S. Federici on adjusting models vs. adjusting

country data (https://doi.org/10.5194/essd-2022-245-CC1). A pragmatic interim solution that we propose is to adjust the global models' results if the analysis is partly or predominantly focused on country or regional levels, and considering adjusting the sum of country data to the models' results if the analysis is focused on climate mitigation efforts at the global level relative to modeled emissions pathways. This approach, followed also by the UNEP Emissions Gap Report 2022 (see Box 2.1 therein) ensures that country estimates are consistent with those reported by countries themselves to the UNFCCC, and that global estimates are consistent with the carbon cycle, scenarios and climate science literature used in the IPCC Assessment Reports. Given the focus of this study on regional and country-level estimates and on disaggregating fluxes into different categories, we here adjusted BMs' results to country data".

A major comment on the method (description). I do not think the method has to be changed, but I think we need a full description of TRENDY. Is there a TRENDY paper that describes the different runs, consequences, etc? We need a list of S0, S1, S2, S3, etc, and a description of what they mean. A figure may be nice. It seems to be so many issues arise because of the way the carbon budget has defined things. What is the land sink in terms of assumptions (S2)? How do the BMs match to the Sx conventions? What are the DGVMs not used for net LUC? How does the LASC fit in? And what does all this mean for the 'gap'? Ok, I can imagine a response from the authors would be that this should appear elsewhere, it should be in the global carbon budget paper, or in a TRENDY specific paper, etc. But, unless one has a 10-year experience with TRENDY and its protocol, understanding some choices made in the paper, and the consequences thereof, is difficult.

Thanks. We have re-written the methods ("Global models") to address all points raised by the reviewer. However, we refer to two original papers (Sitch et al., 2015, and Obermeier et al., 2021) for further details - since we do not use any other simulation than S2 we refrain from explaining the other simulations here. We have instead clarified that we only use the S2 simulation.

Another major comment, which I did not notice to the end, is that the data is just a table? I was expecting something more comprehensive then this. Basically, the paper is providing a summary of key results, not the data used in the paper and for the analysis?

Thanks. We considerably expanded the dataset in the online repository (https://zenodo.org/record/7541525#.Y8WF8ezMJEI), including for each country a time series 2000-2020 for: each BM, with results for each land-use category; the ensemble mean of the DGVMs, with the sink in non-intact / managed forest ( i.e. the adjustment applied to BMs); the NGHGI data for each land use category. Furthermore, in the same repository we made available at two different resolutions the forest map that we used and the detailed protocol to process DGVM data.

I have some more specific comments, building on my points above, in order they appear in the article:

1. General, lines 67+. What is the dataset? It is a dataset which reconciles the different between two datasets? Or is it an additional disaggregation of the TRENDY dataset? Perhaps something like this is needed: "Here we provide additional disaggregation to existing models runs from DGVMs to allow a reconciliation between…".

   Thank you. We rephrased that sentence but added the term "disaggregation" elsewhere in the text.

   I have not looked at the actual dataset, but I am sort of curious of what it actually is. Is it a better version of the TRENDY dataset?

   Yes it is based on Trendy 11, just the disaggregation is different (intact forest, non-intact forest, other). Now it is online and includes also all results from BMs and NGHGIs

It should be noted that we updated the results to make it consistent with the Global Carbon Budget 2022 (at the time of the original submission, only GCB 2021 data were available). While results changed a bit (i.e., now we have a better match between BMs and NGHGIs at the level of LULUCF, but a less good match for forest land and deforestation), the message of the paper does not change.

2. General, dataset: Ok, I have now looked at the dataset, and it is Table S1? Is that it? I was expecting country level estimates at the same level of detail as in the figures. I can't really do much with this dataset, it is only a table with summary statistics?

Thanks. We considerably expanded the online dataset, see above

3. Line 181+: Sland is S2? This assumes the land areas in PI? If the land areas were allowed to evolve over time, as they did in reality, this leads to the LASC but also includes LUC? But the LUC estimate, based on BMs, does not include any indirect effects. The total land sink is S2+BM+LASC? Alternatively, this is S3? LUC according to DGVMs would be S3-S2? To my general comment above, it really needs a figure to explain this, and put a magnitude on some of these effects. According to Friedlingstein et al 2022, "The resulting loss of additional sink capacity amounts to 0.9GtC/yr", like this is ~10% of the total emissions? This is not trivial! DGVMs have the ability to consider evolving land areas and indirect effects on LUC, but those results are ignored? In the context of comparing to NGHGIs, this seems puzzling! DGVMs are uniquely positioned to bridge the gaps, and this is what the authors do, and so what are the implications if BMs are not the starting point and instead DGVMs are?

In response to this and other comments by this and the other reviewers on the use of the TRENDY DGVMs and the loss of additional sink capacity (LASC), we have consolidated our discussion of the LASC and moved this part to the end of the methods section. We have also condensed the explanation, since the key element of our translation between global models and NGHGI - the managed forest fluxes - is not substantially influenced by the loss of additional sink capacity. We thank the reviewer for pointing out to us that we have not clarified sufficiently that this issue is only of marginal importance to our approach and results.

Line 181+: Building on the previous comment. What is the justification for starting with the BMs? One could start from the LUC estimates from DGVMs? That would be more consistent? It would be useful to explain this more, and potentially show how the BMs differ to the DGVMs for LUC? I presume through the Sx runs, it is possible to make a self-consistent definition of the total net land sink, and then disaggregate into LUC and land sink components that are most useful. If the DGVMs are used for LUC, does the gap between UNFCCC and DGVM LUC differ as much as if BMs were used?

We have re-written the methods section on global models to clarify why BMs and not DGVMs are used to quantify the land-use emissions in our study and in the Global Carbon Budgets, and why estimates from the two types of models are not combined. We acknowledge that the methods for all budget terms in the Global Carbon Budgets evolve over the years to reflect the state of the art and we are happy to revisit the reviewer's suggestion once BMs and DGVMs can be usefully combined. Since the purpose of our paper is to provide a pragmatic approach to translate between the two most common assessments of land-use emissions (NGHGI and the estimates from BMs), we believe that a discussion about alternative approaches via DGVMs would not add to the clarity of our study's goal.

4. Line 205: This text sounds like the country-based data are the national submissions? Perhaps "We used the most up-to-date and complete compilation of country-level LULUCF estimates (Grassi et al), … This dataset builds on … [and mention submissions]". Since "submissions"

is used before Grassi et al, it reads as though Grassi et al is the UNFCCC country emissions. Some rewording will avoid confusion.

Thank you, we did follow the suggested text.

5. Line 275: The "but exclude land-use change". Technically, the process is included in the DGVMs, but the area that underwent LUC is excluded (basically, help at PI). Something like "but the land area is held at PI values as a proxy to exclude LUC"? If correct?

The sentence has been deleted

6. Line 275+: The steps "results were first…and then" could be better explained. The "first" step, with Hansen et al, is to ensure consistent definitions of forests? The "then" or "second" step, is to mask managed land by assuming intact forests? The main point is to explain why you do the "first step", as you don't say why (I don't think). There is also an issue of how the PI area maps to the current intact area? Are these all subsets (PI > Han > Intact)?

The paragraph has been slightly rewritten, and the full protocol has been made available in the online repository.

Importantly, we clarified that the Hansen mask "ensures that the current forest area, and not the pre-industrial one used by S2 runs, is applied in this study"

7. Line 280+. Somewhere, perhaps in a separate paragraph, can you define what "intact" forests are, and why they are a good proxy for "managed land". Not obvious, even if the statistics look rather good in the figures.

Thanks. We added "Intact forest is defined as areas without detected signs of human activity via remote sensing (Potapov et al., 2017) (Figure 2a), which a previous study (Grassi et al. 2021) found being a relatively good proxy for "unmanaged" forest in country reports."

8. Line 316: Just clarifying. The managed land mask is frozen at 2013 values, it has no trend? Any thoughts on the potential implications? The 2% is based on the total area, since LULUCF is a change, a 2% change to forest area is rather significant? Basically, it is the changes that are relevant? And 2% is a big change?

Yes, there is no trend in the intact forest area we used. This simplified the analysis. If this approach had impact on the estimate of land-use changes from BM, then 2% in forest area would be relevant. But since it affects only SLAND, the impact is small: it could be roughly assumed that our 2001 SLAND sink is underestimated by 1% and the one in 2020 is overestimated by 1% – well below the uncertainty from DGVMs. We have clarified it in the text.

9. Line 321: The LASC sits on the LUC side? Or you saying that the intact mask has an area less than the PI forest area, so any LASC are masked out? I don't really understand this paragraph, but it is obviously worth mentioning. It probably needs an explanation for people that are not experts on TRENDY protocols and LASC!

In response to this and previous comments (see above) we have consolidated and re-written in a clearer way the explanation of the implications of the LASC.

10. Line 340: What is deforestation here? Is this the process of forest to non-forest, or is this a positive (gross) flux defined in BMs?

Yes, it is forest to non-forest transition, and it is also the positive (gross) flux defined in BMs. We tries to clarify in all occasions that we refer to "fluxes"

11. Line 343+: It is sticks out that H&N is quite different to the others. Can you provide some more explanation on this?

We now used the GCB 2022 values, and now H&N is much closer to the other BMs. The main reasons is that previously H&N allocated shifting agriculture differently than the other BMs. The reasons why we changed from GCB 2021 data to GCB 2022 is that the latter data were not available at the time of our submission (July 2022). Even if the new results would not change the overall message of our study, we think that this change makes sense. Since our manuscript will eventually likely be published in 2023, ignoring the latest data from the GCB 2022 would sound a bit strange.

12. Line 383+: Again, this LASC seems to be something the reader has to have a good handle on to interpret some of these results. You say it is taken care of, which I can trust, but in the methods I think the reader has to have a much better of understanding of LASC, etc, to be able to pass the comments here.

See comment 9.

13. Line 400+: Why is the method worse in Canada, China, India, etc. You touch on this later, but perhaps just add some pointers that you dig into these differences later. As a reader, I am interested in in why things differ, not just stating they differ.

In this part we describe the results for LULUCF, later on we provide more information for each category, trying to add more explanation than before. We have rewritten the text to reflect the changes from GCB 2021 to GCB 2022. Some of the reasons of the differences are difficult and may deserve future studies.

14. Line 534: After reading the size of the LASC, can you be more specific on the "is likely an overestimate". The LASC seems to be not insignificant! This is a constant theme throughout the paper. LASC seems like a rather significant issue, but it is randomly brought up (it is as if a diligent author noticed "oh, don't forget LASC, we should mention that here"). Does the article need a more systematic discussion of LASC?

See comment 9.

15. Line 550+: A "short-term and pragmatic fix". The discussion is really how to marginally modify the status quo, what can BMs tweak and what can UNFCCC tweak? I sort of see more fundamental questions rising. Are BMs, which ignore indirect effects, too outdated now? They were good when Houghton first did it, but hey, we can do better now? Line 558 mentions are fairly fundamental issue of land management and demographic models, but then Line 559 says "a greater disaggregation" of BMs. What is the point of disaggregating BMs if demographic models are needed? One could even think in terms of observational constraints. UNFCCC inventories built on forest inventories is close to an observable, but BMs don't include indirect effects? DGVMs may be more similar to a UNFCCC inventory in terms of processes included (indirect effects), give or take the area issue and annual variability, so could DGVMs be better overall for the LUC (vs BMs)? Or are DGVMs too uncertain? Basically, could a "short-term and pragmatic fix" be to drop the BMs? (sorry to those running BMs). BMs seems like a simple climate model that ignores the carbon cycle? BMs are not really inputs into climate models, that is more land transitions? To rephase your comment, "what is the long-term and non-pragmatic fix"?

See our previous answers that we now clarified in the manuscript why currently BMs are still advantageous to be used and DGVMs only provide ancillary information on the land-use emissions term. We have also rephrased the text to clarify that it is not a "fix" to the BMs that

we meant with the "pragmatic fix", but that we were referring to the translation between NGHGI and BMs.

16. Line 574+: A good point to bring out is how much of the gap is due to differences in areas and how much is due to indirect effects? Quantifying that would be a very valuable exercise, and gets closer to original notion of separating anthropogenic effects? Historically, the UNFCCC approach of managed land is a proxy for anthropogenic. Do we have better science now that we can use a different proxy? Does the data in the paper give a method to better define anthropogenic?

Thanks. We tried to reinforced the text in this section. Overall, we do not see easy alternative to the managed land proxy in the short-medium term. And nor did the authors of the 2019 Revised IPCC Guidelines.

On the area of managed land, we added new text and numbers in the method section:

"At the global level, NGHGIs indicate about 3.7 and 0.7 billion ha of managed and unmanaged forest, respectively. In comparison, the IPCC Special Report on Climate Change and Land (IPCC, 2019b) indicates 2.9 and 0.9 billion ha of "forest managed for timber and other uses" and "forest with minimal human use", respectively. In terms of global ice-free land surface (ca. 13 billion ha), about 75-80% of land is considered to be under some form of human management (Erb et al. 2017, IPCC 2019), with the rest being unmanaged forested and unforested ecosystems (ca. 2 billion ha) or other land (barren, rock). By contrast, the BMs consider a much smaller area of managed forest than NGHGIs (e.g., 1.4 and 1.3 billion ha by BLUE and H&N, respectively). Finally, the areas used in this study - based on the combination of non-intact and intact forest plus country-specific information (for Russia, Canada and Brazil) - are about 3.3 and 0.8 billion ha for managed and unmanaged forest, respectively (Figure 2b, Supplementary Table 1). Australia is the country with the greatest difference between the area of managed forest used in this study (0.04 billion ha) and the NGHGI (0.13 billion ha, although the NGHGI assumes a large part of this area to be in carbon equilibrium)"

17. Line 618: "it is unlikely that countries will change". Bleh. What time frame, short to medium term? Countries routinely make changes, revisions to LULUCF can be rather significant too. I disagree with this starting point. If we (the scientists) but forward good reasons to change, in IPCC reports, maybe they will change. That doesn't mean they change to BMs or DGVMs, but they change. The managed land issue was a poor proxy for anthropogenic, with many issues. If the science has evolved to do better, just say that, and don't prejudge whether countries are rigid or not. Same goes for the science community too... The budget once had H&N, then included BLUE, and now OSCAR also. It used to have a residual sink, now it has DGVMs. We have to think in terms of evolution, not holding onto old ideas.

Thanks. We changed the text as "Overall, in the short term (i.e., before 2030), it is unlikely that countries will change their approach to reporting anthropogenic land-use fluxes from managed lands only, due to methodological reasons (most NGHGIs are based on direct observations, which cannot fully separate human-induced and natural effects) and policy concerns (compliance risks). In addition, any changes would first need to be included in new IPCC guidelines and approved by the UNFCCC, a process that usually takes many years".

We understand the arguments that, if better solutions exists, science should propose them and insist for their application. But right now we do not see viable and pragmatic alternatives for NGHGIs, other than the incremental improvements we suggested in our discussion. In that

regard, models are more flexible than NGHGIs, models are not constrained by IPCC Guidelines, and we therefore suggest that the adjustments be made to models not to the NGHGIs.

18. Line 651: I don't like this sentence. It implies that the science community has done it wrong, and when corrected, land is a net sink overall. One could also write: "When countries include the natural land sink together with source from LUC, they can report a sink". What are you trying to say, perhaps: "When results from DGVMs and BMs are redefined to include an expanded definition of forests and indirect effects as a part of the anthropogenic sink, our analysis confirms that sink estimated in NGHGIs". It is difficult to get the language right, but I think you want to avoid implicitly assuming NGHGIs are superior.

Thanks. We shortened the conclusion and deleted that sentence. But trying to take into account the suggestion elsewhere

19. Conclusions: An important point to emphasis somewhere here is the distinction between climate modelling and inventories. The NGHGIs cannot be fed into a climate model (or integrated assessment model), without those models making potentially significant modifications. Sure, let countries get excited about the sink in their NGHGI, but this has to balanced with a smaller carbon budget, potentially significantly for some countries, and the ability to maintain a sink under climate change, may not be trivial. A major point here is the NGHGI approach is really a poisoned chalice (in my opinion), countries and some scientists have yet to realise that!

Thanks. We extended the discussion on the risks for countries associated with large managed area. On the smaller C budget , we see your point and agree, but also feel that it is a bit outside the scope of this paper (it hase more broadly discussed in Grassi et al. 2021). Nice analogy the 'poisoned chalice' !

20. Figure 1: Can you make a similar figure, but for S0, S1, S2, S3, LASC, BMs, NGHGIs? Not sure if possible, but consider it a challenge!

Since we have re-written the text to focus on S2 (see previous comments) we do not see the need to provide more detail on other TRENDY simulations or the LASC.

21. Figure 3: I guess the average as a median would just be the middle model? In any case, the "gap" to reconcile is really a heavy function of the BM. If OSCAR was correct the gap would be much smaller, H&N much larger. What are these BMs doing differently?

We now use the GCB 2022 data (see above).

On the mean/median, we use the mean and not the median exactly because of the issue mentioned by the reviewer.

The main difference between BMs is probably the input data (we have now added this text in the methods), as discussed in Friedlingstein et al. 2022 (and now added in our text):

- BLUE is based on LUH2

- H&N is based on FAO data

- OSCAR uses both LUH2 and FAO data to constrain the model runs

And when they are compared to NGHGIs, it really depends on what you look at. The gap for OSCAR is smallest for forest, but largest for deforestation. This might again hint to a different separation of fluxes from shifting cultivation in OSCAR.

22. Figure 5: I guess a is Sland, disaggregated? For b, at what stage do you filter out models. Is there reason to believe the model at -2 and constant? Is there a trend in Sland, and is that trend the same / different as the trend in non-intact? How does the LASC fit into this figure?

Yes, it is SLAND, we now added it. We take all the 16 DGVMs used in GCB 2022, without filtering.

We do not perform a trend analysis since our translation between global models and NGHGI does not require statistical measures on the trend.

On the LASC, please see the previous comments.

23. Figure 6: Why is H&N doing so well?

We now use the GCB 2022 data (see above), and H&N changed

24. Figure 10: Potentially useful figure. It would be even better if the LASC could be included? That would be a nice visual of the size of this effect and its potential importance. In the same vein, perhaps including LUC from the DGVMs?

Thank you for the suggestion. Since, as explained in previous responses, we now clarified that the LASC does not play a major role for the key element of the NGHGI-BM mapping (the managed forest fluxes) we do not think adding the LASC to Fig. 10 will be helpful to the reader.